# Willingness to help climate migrants: A survey experiment in the Korail slum of Dhaka, Bangladesh

**Rachel Castellano**[1], **Nives Dolšak**[2], **Aseem Prakash**[1]*

**1** Department of Political Science, University of Washington, Seattle, Washington, United States of America, **2** School of Marine and Environmental Affairs, University of Washington, Seattle, Washington, United States of America

* aseem@uw.edu

**Data Availability Statement:** Relevant data has been uploaded to the Harvard Dataverse (https://dataverse.harvard.edu) with the following DOI: https://doi.org/10.7910/DVN/HTV3DT.

## Abstract

Bangladesh faces a severe rural to urban migration challenge, which is accentuated by climate change and the Rohingya crisis. These migrants often reside in urban slums and struggle to access public services, which are already short in supply for existing slum dwellers. Given the inadequacy of governmental efforts, nonprofits have assumed responsibility for providing essential services such as housing, healthcare, and education. Would local slum-dwellers in Dhaka be willing to support such nonprofits financially? We deploy an in-person survey experiment with three frames (generic migrants, climate migrants, and religiously persecuted Rohingya migrants) to assess Dhaka slum-dwellers' willingness to support a humanitarian charity that provides healthcare services to migrants. Bangladesh is noted as a climate change hotspot and its government is vocal about the climate issue in international forums. While we expected this to translate into public support for climate migrants, we find respondents are 16% *less likely* to support climate migrants in relation to the generic migrants. However, consistent with the government's hostility towards Rohingya, we find that respondents are 9% *less likely* to support a charity focused on helping Rohingya migrants. Our results are robust even when we examine subpopulations such as recent arrivals in Dhaka and those who have experienced floods (both of which could be expected to be more sympathetic to climate migrants), as well as those who regularly follow the news (and hence are well informed about the climate and the Rohingya crisis).

## Introduction

Climate change is an important global policy issue. Increasingly, leading policymakers, business leaders, celebrities, and non-governmental organizations emphasize the need for quick and substantial efforts to tackle the crisis. While policy changes such as a transition from coal to renewables in electricity generation are critical, such profound changes will eventually require citizen cooperation as well. This holds for policies targeted at climate mitigation and climate adaptation. The former pertains to policies to reduce the emissions of greenhouse

**Funding:** The authors received no specific funding for this work.

**Competing interests:** The authors have declared that no competing interests exist.

gases, while the latter pertains to policies that increase the resilience to or protect from the effect of climate change [1, 2].

To what extent are citizens in a developing country willing to expend private resources to support an important climate adaptation policy, namely climate migration? Climate change is increasing the severity and frequency of extreme weather events. Because this is making some areas unfit for human habitation, individuals could adapt to climate change by migrating to a more hospitable area [3]. There is a rich literature examining support for climate mitigation policies such as carbon taxes and cap and trade, especially in developed countries [4–6]. This is among the first papers to examine public support for non-governmental organizations that work on climate adaptation by providing public services to climate migrants. We focus on Bangladesh, which is among the world's most densely populated countries, and is often identified as a climate change hotspot [7]. It faces risks from rising sea-level, increased frequency of floods and droughts, and salt-water intrusion [8].

Theoretically, our paper speaks to the broader debate on citizen perceptions of salient global issues and how they form an opinion about actors, both governmental and non-governmental, that work domestically on these issues [9]. We offer several different perspectives on why respondents might or might not be willing to support an organization that supports migrants, such as empathy-driven giving and competition over scarce services. Thus, we do not have a theoretical position on which perspective will prevail and address this question empirically.

International treaties obligate domestic governments to translate them in domestic policies and enforce them. A government's willingness to enforce international treaties as domestic policies depends, in part, on domestic support for these policies. However, citizens are unlikely to support policies that they view as an elite imposition because these policies do not address their concerns or sometimes militate against their core beliefs, as in the issue of gender equality [10] or same-sex marriage [11]. Governments fear high political costs when citizens believe that new policies clash against their interests and beliefs [12]. In some countries, international trade agreements are also viewed as elite impositions that enrich multinational corporations at the expense of workers [13]. Broadly, the populist rhetoric against globalization falls in this category. Climate change is an important global issue, but policies such as carbon taxes have invited populist backlash even in developed countries (such as the "yellow vest" protests [14] or the defeat of two carbon tax initiatives in the state of Washington [15]).

In addition to the concern about elite impositions, there is an emerging literature in development studies on "democracy recession." In the last two decades, there has been a massive crackdown against NGOs worldwide. Governments have incentives to crack down on foreign aid when they perceive NGOs are working with their political opponents and when they perceive that NGOs do not have citizen support (therefore, the political costs of cracking down are low) [16]. Sometimes citizens believe, often abetted by autocratic governments that control the media, that NGOs work for western agendas instead of local concerns. Scholars term this as the NGOization of civil society [17]. The literature noted that as foreign donors route aid through NGOs as opposed to local governments, NGOs became visible in public service delivery–sometimes even more than the local government [18]. For example, NGOs flooded Haiti after the 2010 earthquake. Not surprisingly, Haiti has subsequently acquired the label of the "Republic of NGOs". Competition among NGOs for funding meant that NGOs were perceived as working on agendas dictated by their donors [19]. Thus, citizens sometimes become wary of even local humanitarian NGOs, especially when they work on "global" agendas.

The extent to which support in developing countries for climate action measures up with international concern is unclear, especially if it involves citizens incurring private costs. Moreover, while the threats of the climate crisis are visible, most developing countries do not have the resources to address the climate challenge. Given the level of poverty and other pressing

needs, it is unclear whether citizens in developing countries view climate change as their top policy priority. If citizens perceive climate change as an elite "western" issue, their lack of support could spill over to even non-governmental climate action. In the context of Bangladesh, this paper examines citizen support (in terms of willingness to incur private costs) for a charitable organization that serves climate migrants.

Climate migration is a form of *ex situ* adaptation [20]. Riguad et al. estimate that by 2050, the number of climate migrants in Latin America, sub-Saharan Africa, and Southeast Asia alone will reach 143 million, and that environmental migration in Bangladesh will outpace other internal migrations. Under the pessimistic reference scenario, they predict that 13.3 million people will become climate migrants by 2050 [21]. The National Geographic declared that while Bangladesh is "already grappling with the Rohingya crisis, it now faces a devastating migration problem as hundreds of thousands face an impossible choice between battered coastlines and urban slums" [22]. Scholars expect large-scale migration from Bangladesh's coastal areas to its capital city Dhaka [23]. This poses a policy challenge because Dhaka is already overcrowded, with a population of 18 million that is expected to increase to about 50 million by 2050. Dhaka is the most densely populated city in the world [24], and the living conditions in Dhaka slums are getting worse as about 2,000 people move to Dhaka every day [22, 24].

New migrants require substantial private assistance, given the government's widespread failure to provide basic public services [25]. Family networks certainly help but given the widespread poverty, this help is often inadequate. Consequently, local charities have stepped in [26], often mobilizing substantial funds from the local community. We assess individuals' willingness to contribute to a (fictitious) charity, Bengal Humanitarian Organization, that provides healthcare to migrants. We expect a higher level of support for a charity that serves climate migrants (in relation to generic migrants) given the global advocacy of the climate problem by the Bangladesh government. The local media also reports high levels of concerns in international forums about climate issues. If local residents take cues from the global discourse, we should expect to see higher support levels for climate migrants.

In contrast to climate migrants, we expect a lower level of support (in relation to generic migrants) for a charity that serves Rohingyas, refugees from neighboring Myanmar. While there is widespread global sympathy for Rohingya refugees, the Bangladesh government treats them harshly, and the local media portrays them negatively, often blaming them for rising local crimes. In international forums, Bangladesh demands quick repatriation of the migrants to Myanmar. As we explain further in our Methodology section, we choose to use the term 'religiously persecuted migrants' in our survey experiment to elicit responses about the Rohingya because of the emotional saliency the term Rohingya carries in Bangladesh.

Our findings are mixed. As per our expectations, we find lower support (9% lower than the reference group) for a charity that serves Rohingyas. Much to our surprise, we find a lower willingness (16% lower than the reference group) to support climate migrants as well. Our results are robust even when we examine subpopulations such as recent arrivals in Dhaka and those who have experienced floods (both of which could be expected to be more sympathetic to climate migrants), as well as those who regularly follow the news (and hence are well informed about the climate crisis).

## Migration and climate change

An alarming increase in climate-related natural disasters is leading to population dislocation. Consequently, policymakers increasingly recognize the emerging challenge of climate migration. While developed countries are responsible for the bulk of accumulated emissions driving

climate change, developing countries, particularly in the global south, are disproportionately affected by climate change and are already experiencing large-scale climate migration [27–29]. The majority of climate-induced displacement is typically internal to the migrant's home country, though cross border climate migration is also expected to increase [30].

A key debate around climate migration with important theoretical and political implications is about who counts as a climate migrant. Biermann and Boas [31] argue against subsuming 'climate refugee' under the 1951 Geneva Convention Relating to the Status of Refugees. Instead, they advocate for a new international framework dedicated to the specific needs of climate refugees. Betts [32] argues for creating a category of 'survival migrants', defined as those who move outside their country of origin for threats to which there is no domestic remedy. Drawing on the experiences of climate-induced displacement in the Pacific Island of Tuvalu, some scholars reject the image and discourse of climate refugees [33, 34] because it is politically charged.

We also recognize that the term "climate migrant" is problematic since it could emphasize the "pull" of the destination more than the "push" of the source region as the driver of human movement. In addition to the negative connotations, this could also reduce the implied responsibility of the international community for their welfare. Indeed, The International Organization for Migration encourages the use of the term 'environmental migrant' defined as:

"A person or group(s) of persons who, predominantly for reasons of sudden or progressive changes in the environment that adversely affect their lives or living conditions, are forced to leave their places of habitual residence, or choose to do so, either temporarily or permanently, and who move within or outside their country of origin or habitual residence [35]." Due to our focus on climate change, and the contested nature of the climate refugee label, we use the term climate change migrant in this study, while recognizing its limitations.

Bangladesh is identified among the first countries to face the consequences of climate change, including migration [30, 36, 37], and ND-GAIN Country Index ranks Bangladesh as the 20th most vulnerable to climate change among 181 ranked countries [38]. About 40% of Bangladesh's land area and 46% of its population are located in the Low Elevation Coastal Zone areas that are between 1 to 20 meters above sea level [39]. In fact, as per Raigud et al., a one-meter rise in sea level is estimated to result in a loss of more than 4,800 square kilometers of land area [21]. Because the Bangladesh government faces resource problems in constructing the "hard" adaptation infrastructure, such as seawalls, migration could be viewed as an individual-level climate adaptation strategy [40–42]. Hassani-Mahmooei and Parris [37] predict changes in migration from the west, which is drought-prone, and the south, which is vulnerable to cyclones and floods, towards the northern and eastern regions. Their model predicts between 3 and 10 million internal migrants over the next 40 years in Bangladesh.

Historically, there is a steady stream of rural migrants relocating to cities in search of livelihood [43, 44], particularly in Dhaka. Climate migration is a continuation of an existing trend of rural-urban population movement. Newly arrived migrants require basic public services such as healthcare. However, governmental resources are already stretched thin with existing obligations. As scholars have noted, nonprofits often emerge to correct governmental failures in public service delivery [45]. While nonprofits secure funds from various sources, local nonprofits often rely on local funding. In this Tocquevilian [46] model of local level voluntary action, nonprofits raise resources from the communities they serve. Further, recent work suggests that climate migrants might be perceived differently from other migrants. In the context of Germany, Helbling (2020) reports that German respondents are more supportive of climate change migrants, in relation to economic migrants [47]. Hence, we examine whether Dhaka's

slum dwellers are willing to contribute to healthcare services for climate migrants who have joined their community.

Poverty is not always a barrier to philanthropy. As a percentage of income, the poor donate more to charities than the rich [48, 49]. In the United States, those in the top 20 percent of incomes contribute, on average, 1.3 percent of their income to charity while the bottom 20 percent donated 3.2 percent of their income [50]. The reason may be that empathy often drives charitable giving [51].

However, migrant reception by local communities is complex. Weber and Peek [52] report that while there was a general warm and compassionate reception of Hurricane Katrina evacuees, community leaders expressed concern that evacuees were moving ahead of local people in need of public assistance on lengthy waitlists. Ishtiaque and Mahmus [53] find that rural-urban migrants primarily move to Dhaka to access the informal economy, find a job, or earn money, and 70% of respondents believed that their migration objectives had been fulfilled. This inevitably results in increased competition for resources, particularly in areas that already face resource scarcities. Dhaka slums are overcrowded and lack adequate public services, such as housing and health [25]. Thus, this study contributes to the literature on the reception of different types of migrants among communities that are already experiencing economic struggles.

Bangladesh was the first South Asian country to formulate a Climate Change Strategy and Action Plan. In 2011, climate protection was given a stronger legal status by an amendment to the constitution, although its impact on domestic policy remains unclear. In recognition of Bangladesh's climate leadership, Prime Minister Sheikh Hasina was awarded the 2015 United Nations Champions of the Earth award. Given the extensive focus on climate change in media and the strong advocacy by the Bangladesh government in global forums, we hypothesize:

H1: Survey respondents will be more willing to support climate migrants in relation to generic migrants.

We also test for public support in the context of another migration crisis that Bangladesh is facing: Rohingya who have fled neighboring Myanmar due to religious persecution. This issue has gained considerable international attention. Myanmar leader and Nobel Laureate Aung San Suu Kyi appeared before the International Court of Justice in The Hague to defend her country against the charge of genocide. However, the regional politics of the issue are complex. Although Rohingyas share the Islamic faith with most Bangladeshis, Rohingya have not been well received in Bangladesh. Ullah [54] highlights the systematic brutality towards the Rohingya population, which spans decades in Myanmar and Bangladesh. For the domestic audience, the Bangladesh government often portrays Rohingyas negatively, highlighting their criminality and illegality. The government seeks to confine them in camps, located around the Cox Bazaar area. It is very keen to repatriate them back to Myanmar; indeed, recently, it even cut off mobile phone connections to these camps [55]. In addition to the law and order issue, citizens fear that Rohingya refugees' cheap labor depresses wages in the local job market [56]. The government is also starting to implement its plan on relocating Rohingya to an island called Bhashan Char, off the southern coast of Bangladesh. This is an incredibly controversial decision because of its vulnerability to cyclones [57]. Because of these negative narratives about Rohingyas, we hypothesize:

H2: Respondents will be less willing to support Rohingyas in relation to generic migrants.

## Methodology

We focus on the charitable giving of slum dwellers, who constitute the majority of the Bangladeshi population and compete with new migrants for valuable public and private resources.

Hence, their willingness to donate to healthcare services for new migrants sets a high bar for us to assess the level of domestic support for climate issues. After receiving permission from the University of Washington's Human Subject Division (IRB ID: STUDY00009013), we interviewed (over Skype) several well-established survey firms in Bangladesh. We hired Sustainability Services Limited, located in Dhaka, and compensated them for administering the survey. We informed them about the ethics guidelines, including respect for the local law as well as the issue of prior, informed consent. Consequently, all respondents were adults (18+) and their verbal consent was taken before administering the survey. The payment to this firm was facilitated through University of Washington.

With the survey firm, we discussed in length about the sampling strategy and survey methodology (including sending women surveyors to interview female respondents, given the traditional nature of the Bangladesh society). We also consulted the survey firm to ensure that the survey (see S1 Appendix) in the Bengali language was both culturally appropriate and informative. For example, we had an extensive discussion on the appropriate name for the charity and what amount we should ask for in the question about donating. The firm managers also encouraged us to employ the phrase persecuted minority instead of Rohingyas in the survey instrument because the phrase Rohingya is extremely volatile in Bangladesh. Thus, while the persecuted minority clearly signals that we are asking about Rohingya, it will not unleash an emotional reaction from the respondent.

We recognize that Buddhists and Hindus could also be considered persecuted minorities in Bangladesh (although the prevalence of this persecution has decreased under the current Awami League regime) and we raised the issue with the survey firm in Bangladesh. We were advised that Hindus and Buddhist tend not to migrate to Dhaka but instead migrate to other places, like India. Furthermore, there are no media reports of large-scale violence against Hindus and Buddhists under the current Awami League regime. Indeed, this regime has cracked down on Islamic fundamentalist groups that worked with Pakistani Army during the Liberation war and were often in the forefront of fomenting violence against minorities. Thus, to guard against any confusion on the nature of religiously persecuted minorities, we chose the language in the treatment frame carefully: "religious violence is causing a large displacement of people." Hence, we are confident that respondents interpret the term "persecuted minority" as referring to Rohingyas.

We first piloted the survey with about 200 participants to ensure that our questions were clearly understood. Then, the survey firm conducted a 1,800 in-person survey of individuals, exposing them to three different frames describing a fictitious charity's work. Our firm administered the survey in the Korail slum in Dhaka. The survey team collected data from almost the entire Korail slum. They started with identifying five blocks based on the scoping study. The entire slum was then grouped into 20 clusters based on these blocks. Employing a single-stage cluster sampling considering gender, religion and occupations, the team interviewed 100 respondents in each cluster. When respondents did not give their consent to take part in the interview, the survey team moved to another respondent. Only one household member in each family was interviewed in this study.

Given that Dhaka has more than 3,300 slums, we recognize the issue of generalizability. These slums differ on many aspects, including the percentage of slum dwellers receiving medical services from NGOs (47% in Korail) and the composition of slum population in terms of areas/regions they come from. Based on our extensive discussion with the survey firm, we decided that given the heterogeneity among slums on different dimensions, Korail provided an appropriate survey site. However, we hope that our paper will motivate additional work in different sites to empirically assess the generalizability of our findings. Further, our regression

**Table 1. Experimental frames.**

|  | Charity Recipient | | |
|---|---|---|---|
|  | Charity provides healthcare to migrants | Charity provides healthcare to climate migrants | Charity provides healthcare to religiously persecuted migrants |
| Generic Group | X |  |  |
| Climate Frame |  | X |  |
| Persecuted Minority (Rohingya) Frame |  |  | X |

analysis does control for some issues such as prior experience with extreme weather events, a dimension on which the composition of slum populations might differ.

Among the respondents, 97.3 percent identified at Muslim, 2.5 percent identified as Hindu, and 0.2 identified as Christian (see S6 Appendix for a table on demographics of survey participants). The national averages are 89.1 percent Muslim, 10 percent Hindu and 0.9 percent other (including Buddhist and Christian). We have a slightly higher representation of Muslims. This makes our estimates more conservative because Muslims could be expected to be more sympathetic to their co-religionists, Rohingyas, who are facing religious persecution in the neighboring country.

Our sample was equally split among men and women which approximates the national average. 46.7 percent were employed, 21.5 percent were homemakers, and 12.2 percent were unemployed but looking for work. The national unemployment rate is much lower at about 4.4 percent, further supporting the claim about the lack of economic opportunities in Dhaka slums [58].

The survey experiment follows a between-subjects design, where individuals were randomly assigned to one of the three groups (see Table 1). The groups were asked for their willingness to donate to a fictitious charity, Bengal Humanitarian Organization, which provides healthcare to migrants. Depending on the group, respondents were told that the Bengal Humanitarian Organization provides healthcare to migrants generally, climate migrants, or religiously persecuted migrants.

Migrants have different characteristics. The vast literature on migration studies has examined public support when specific characteristics of the emigrant such as religion, gender, skill level, etc. are highlighted. We contribute to this literature by focusing attention to a specific characteristic that the literature has overlooked: climate change as a migration driver. Thus, the generic frame does not highlight any migration driver unlike the two treatment frames. Consequently, this research design allows us to assess the change in public support when one specific migrant characteristic (migration driver: climate change or religious persecution) is highlighted in the two treatment frames while all other information remains the same as the generic frame. This is also why we do not have any open-ended questions to investigate what types of migrants the respondents had in mind after reading the generic frame because we are not examining how respondents perceive generic migrants. Rather, we want to see how support for migrants might shift (in relation to the generic migrant) when one specific migration driver is highlighted.

To further elaborate, the generic migrant frame in our survey experiment is intended to capture migrants who move because of any reason including economic and/or educational opportunities. Thus, in the generic frame, the driver of the migration is not identified. In contrast, in the treatment frame, the migration driver is identified. While there is potential overlap between the generic frame and the other two frames, the objective of the generic category is to provide a benchmark (or reference category) to assess if the willingness to support the charity

changes when a specific migration driver is identified in the treatment frames. Thus, in our survey experiment, frames are identical, except for one factor—the information about the migration driver. Therefore, they are not mutually exclusive. If the migration driver does matter (because it generates empathy or fear) in generating public support, then it has important policy implications.

Surveyors read a brief summary of the charity to respondents before asking them if they would be willing to donate 100 takas (the local currency) to the Bengal Humanitarian Organization. As per Mahumud et al. [59], on average, Bangladeshi households spend $1.4 per month on medicines, which amount to about 120 takas. Based on the advice of the survey company, we rounded it off to 100 takas.

To ensure that respondents understood (and were attentive to) the questions, we then asked them a set of comprehension questions. We limit this analysis to only those respondents (1,443 of the 1,800) who correctly answered all the three comprehension questions. Of the individuals who were excluded, 118 were in the generic group, 161 received the climate treatment, and 77 received the Rohingya treatment (our results hold when we examine the full sample, as shown in in Fig 2). Finally, the surveys asked questions about demographic information, media consumption, crime in Bangladesh, time spent in Dhaka, and experiences with floods.

Our dependent variable, willingness to donate, is a five-level scale (no, probably not, maybe, probably yes, and yes). We discussed this possibility of using some sort of a slider scale to ask respondents for their support for 100-taka donation on say 1–5 scale. Because this was an in-person survey (as opposed to an online one), the survey firm thought that the logistical issues will be difficult if we were to ask respondents to use the slider on the smartphone and might even be distracting. Hence, we decided to work with a 5-point Likert scale–which is consistent with most survey experiments in the climate policy field. In S2 Appendix, we also provide an OLS estimation where we treat the dependent variable as continuous as a robustness check. Our results about the lower support for both climate migrants and Rohingya (with the generic group as the reference category) remain unchanged.

Given the categorical and ordered nature of the dependent variable, we estimate ordered probit models using the full scale. We combine the predicted probabilities of donating (combining "probably yes" and "yes", as well as "probably no" and "no") in post-estimation simulations [60, 61]. Following Dolšak et al., [62], we only combine the predicted probabilities of donating in post-estimation simulations in order to avoid losing precision in our results (as would occur if we collapsed the scale prior to estimation). Our ordered probit results are sample average treatment effects (SATEs), which average the expected percentage change of respondents offering support [62]. Because ordered probit coefficients are on a log-odds scale, they are more difficult to interpret then coefficients in a linear regression. Therefore, we run a simulation 10,000 times to obtain first differences between predicted values. This method requires the construction of alternative scenarios and is more useful for interpreting log-odds than trying to calculate odds or odds ratios [63].

## Results

Among all respondents, 75 percent answered "probably yes" or "yes" to whether they were willing to donate. For the remainder of this paper, we report the results for willingness to donate by combining the "probably yes" and "yes" categories. 86 percent of respondents in the generic group were willing to donate. However, 61 percent of respondents receiving the climate change migrant frame were willing to donate, while 77 percent of the respondents receiving the persecuted migrant frame were willing to donate. Despite the discrepancies between

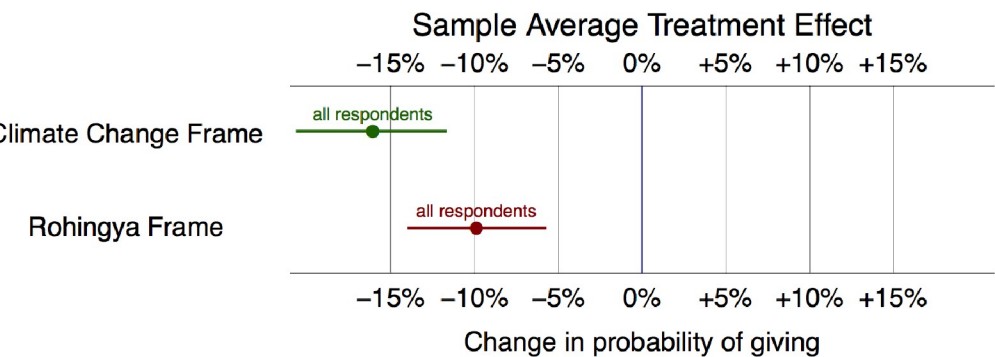

**Fig 1. Differences in willingness to donate between the generic frame and both experimental frames, all else equal (Attentive sample only).** Note: This plot shows the change in the willingness to give for each frame. The generic frame is the baseline (0%). 95% confidence intervals also shown.

these numbers, it appears that survey respondents were very generous. Even though many live in poverty, a high percentage were willing to donate to a charity they had never heard of before. While these results are encouraging, it is possible that our results suffer from social desirability bias, one of the most common biases in survey research [64]. In this case, we suspect that social desirability bias manifests in the respondent's desire to appear charitable. We hope that future research will test this bias by observing what individuals say they will give to migrants compared to what they will actually give.

Fig 1 shows the estimated average effect of both climate and Rohingya frames on our sample of respondents. Much to our astonishment, the data not only fail to support our hypothesis of higher support for climate migrants (H1) but instead indicate *lower* support. As the figure below shows, the probability of giving to the climate migrants is about 16 percent *less* than the probability of giving to the generic group. We speculate that this finding could reflect the disconnect between elite discourse and grassroots perceptions about the importance of climate migration. Further, respondents might view that because climate migration is a "western" issue, they might assume that migrants are probably receiving help from rich international actors. After all, the Bangladesh government is vocally asking for international assistance for climate change.

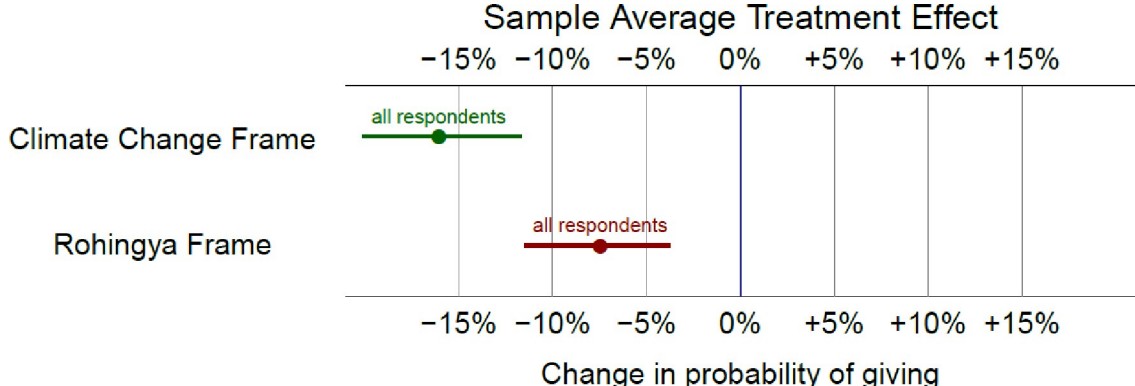

**Fig 2. Differences in willingness to donate between the generic frame and both experimental frames, all else equal (Full sample).** Note: This plot shows the change in the willingness to give for each frame. The generic frame is the baseline (0%). 95% confidence intervals also shown.

Alternatively, Bangladesh citizens might harbor some sort of skepticism about the anthropogenic nature of the climate crisis. For example, a survey conducted in Bangladesh reports that 52% of respondents (and 93% of Muslims in the study) believe that climate change is due to sinful activities and the wish of God [65]–which also implies that any help rendered to climate migrants goes against the wishes of God. While we do not have the data to arbitrate among different explanations for a decreased support for climate migrants in relation to generic migrants, our results are worrisome because grassroots perceptions are particularly important when mobilizing political action around climate change [66].

Another explanation for this surprising result could be the threat of economic competition. It is possible that our respondents might conceptualize climate migrants as permanent residents who will not be able to return home, while conceptualizing generic migrants as seasonal or temporary. Therefore, the opposition to climate migrants in relation to generic migrants could be because of the potential economic threats they pose to a community where resources are already scarce. Indeed, in their study of urban-rural migrants in India, Gaikwad and and Nellis [67] find that city residents belonging to the majority religious group (in the Indian case, Hindus) do not discriminate again rural-to-urban migrants based on religious profiles. Instead, they appear to care mainly about the economic impact of migration.

In line with our expectations, as Fig 1 shows, we find support for Hypothesis 2 that the Rohingya frame will elicit less support than the generic migrant frame. The probability of giving to Rohingyas is about 9 percent less than the probability of giving to a generic migrant. This is in line with our theory that the media and the Bangladesh government have perpetuated harmful narratives about these migrants, resulting in hostility among Bangladeshis.

The results for the sample average treatment and interaction effects discussed below and are provided in tabular form in S4 Appendix.

Our results hold when we include the full sample as respondents as well. As Fig 2 shows, both the Rohingya frame and the climate change frame elicit less support than the reference group (see S5 Appendix for interaction results in tabular form).

## Sub population analysis

### News consumption

The news media might shape opinions about new migrants (see Question 15 in S1 Appendix). Because the Rohingya are portrayed negatively while climate change is deemed a worthy issue, those with higher media exposure levels might show more support for climate migrants and less for Rohingyas. We do not find support for the conditioning effect of media consumption. The interaction between a respondent's frame and their answers about media does not alter the basic results and is consistent across news sub-categories (Fig 3). We suspect that this may be the case because if media is saturated with negative stories, then media might not have a conditioning effect on willingness to give.

Our findings raise a broader issue of how individuals access information on policy issues.

In this survey, we did not ask directly about how respondents learn about policy issues in general or about government's positions on climate migrants. This is for two reasons. First, people typically get much of their policy information from different types of mass and social media. Of course, we do not know if this information is authoritative and if they comprehend this information. Indeed, we are not making any claim on how well informed or poorly informed individuals are about climate change, migration, or any national policies. We recognize that as boundedly rational actors, individuals develop opinions about issues based on incomplete information. Yet, no matter how incomplete or poorly informed individuals are, public opinion matters. And this is what we examine in the context for public support for a

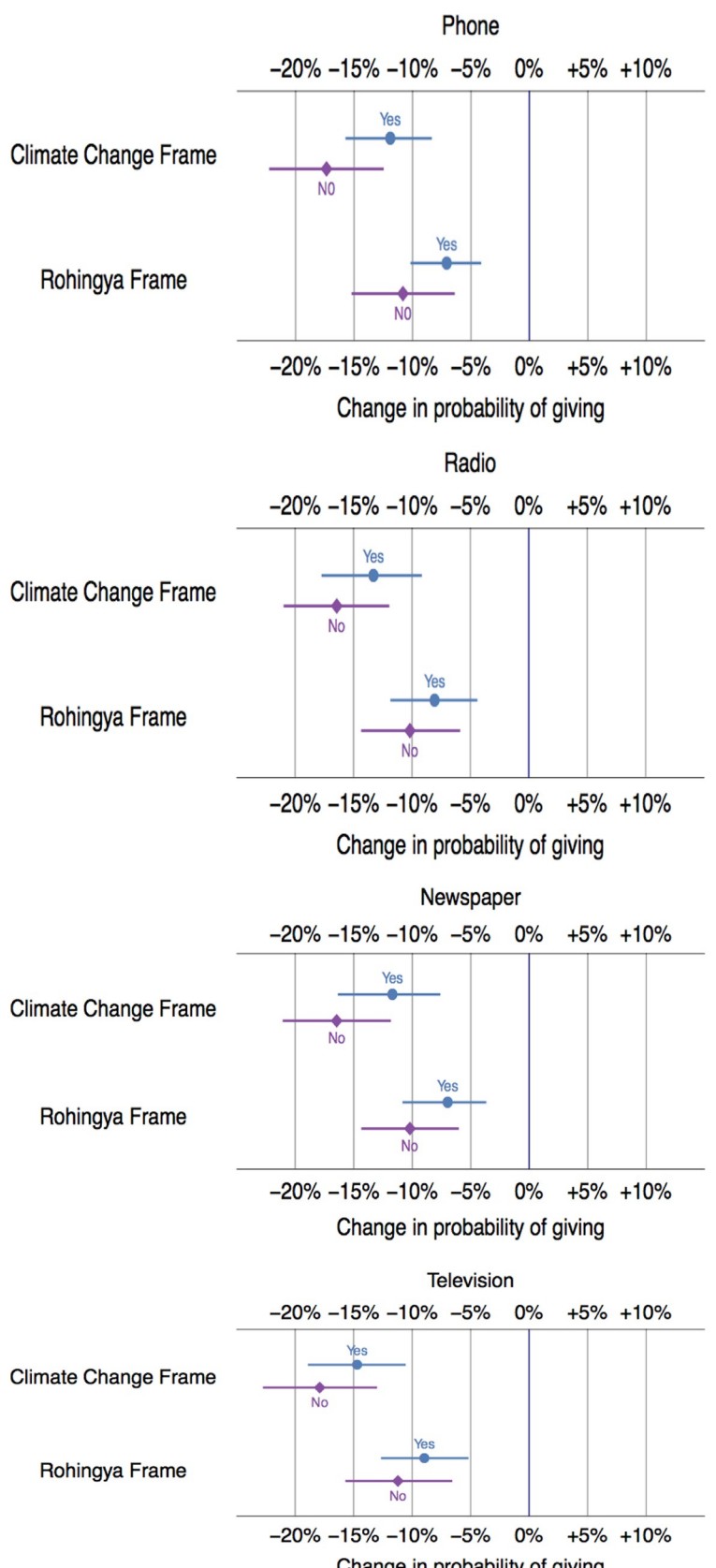

**Fig 3. Predicted differences in willingness to donate based on news consumption within the last 24 hours, all else equal.** Note: These points display the stimulated average effect of both treatments. For example, respondents who received the climate change migrant treatment are 11.7 percent less likely to donate if they had read the news in a newspaper and 16.4 less likely than the generic group to donate if they had not read the news in a newspaper. 95% confidence intervals are also included as horizontal lines.

charity that provides health services to climate migrants and Rohingyas (in relation to generic migrants).

Because the media plays a crucial role in shaping public opinion, we asked questions about each respondent's exposure to different types of mass media (Q15 of the survey) and analyzed if exposure to different mass media might influence respondent's support for climate migrants or Rohingyas in relation to generic migrants. We have not subsumed all types of media into one category because some respondents might rely on newspapers to access policy information, while others might rely on radio. In addition, it is possible that some media might cover climate issues more extensively or effectively than others. For example, television might provide more content about challenges faced by climate migrants in relation to radio. Or, the television footage might create more empathy for climate migrants. If so, those with higher exposure to television might reveal higher support for the charity supporting climate migrants. Our model does control for factors such as prior experience with floods, which might make them more prone to access or pay attention to specific type of climate information. Finally, because this is a survey experiment in which respondents are randomly assigned to different frames, unobserved heterogeneity in respondents' characteristics should not influence support for any frame.

When we rerun the analysis for each media type separately, our results do not change (i.e. support for climate migrants or Rohingyas in relation to the generic frame). This gives us additional confidence that the medium through which respondents might receive information is not changing the support for climate migrants in relation to generic migrants.

## Experience with floods

Might respondents with similar life experiences be more supportive of climate migrants? The "linked-fate" theory [68] suggests that individuals tend to help fellow community members with whom they share life experiences. Because climate migrants of Bangladesh are often escaping rising sea level, the willingness to support climate migrants depends on whether respondents had experienced floods themselves (see Question 17 in S1 Appendix). After all, those who have experienced a natural disaster might have more empathy [50] for those who have had to suffer it as well. Though we can expect respondents to appreciate the monsoon season for replenishing water supplies and help farmer, climate change is likely to accentuate the frequency and severity of even regularly occurring weather events such as the annual flooding. Here as well, as presented in Fig 4, our results remain unchanged.

## Recent migrants

Finally, to further explore the empathy argument, we asked respondents how long they have lived in Dhaka because there might be a difference between those who had recently migrated to the city and those who have lived there for a longer time (15 years). Arguably, recent arrivals might also have stronger connections with their relatives in villages and, therefore, show a higher level of empathy for climate migrants because of the increase in environmental degradation in Bangladesh's rural areas. However, similar to other interaction terms, respondents

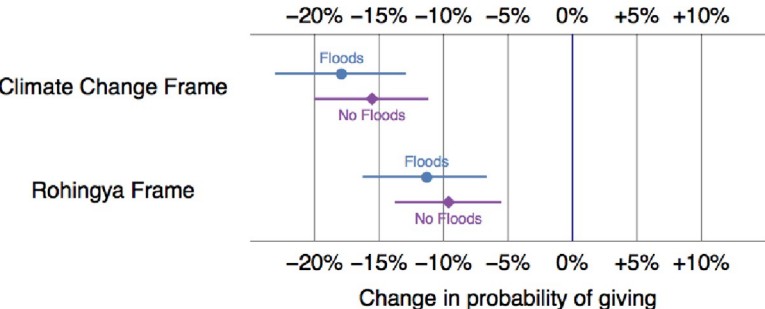

**Fig 4. Predicted differences in willingness to donate based on experiencing floods, all else equal.** Note: These points display the estimated average effect of both treatments. For example, respondents who received the Rohingya treatment are 11.2 percent less likely to donate if they experienced floods in the last year and 9.6 percent less likely to donate if they did not experience floods in the last year. 95% confidence intervals are also included as horizontal lines.

are still much less likely to support climate refugees (and Rohingyas) in relation to generic migrants (Fig 5).

## Conclusion

We expected that Bangladesh citizens will support NGOs providing humanitarian services to climate migrants. After all, Bangladesh is directly impacted by climate change. Hence, our survey findings are contrary to our theoretical expectations. There could be several reasons. First, we suggest that perhaps this is because citizens have already formed their opinion about climate change and NGOs working on this issue. Given the media publicity on climate issues and the constant refrain about its global implications, citizens may feel that it is an elite issue or that NGOs have foreign funding. Second, respondents might believe that because climate change is a global issue caused predominantly by developed countries, the developed North should bear the cost of helping climate migrants, as opposed to citizens of developing countries. Third, the lower rate of willingness to give among climate change migrants in relation to the generic migrants could be related to perceived economic threats about permanent versus temporary migrants. Because we do not specifically explore the reasons for distrust in NGOs, we hope future work will explore this issue in greater detail.

The subject of public support for climate migration (in relation to other types of migration) is relatively new in the climate policy literature (although as we point out in the paper, there is

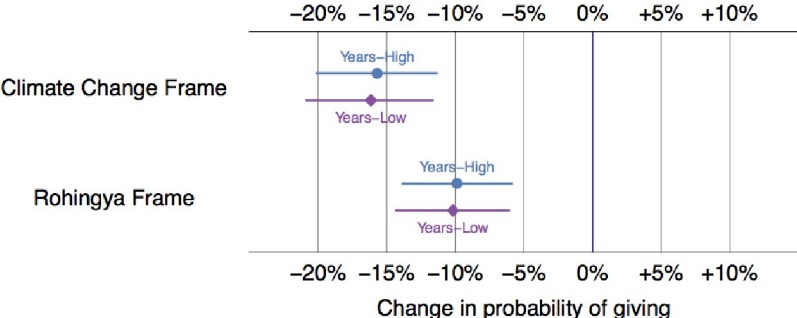

**Fig 5. Predicted differences in willingness to donate based on years lived in Dhaka, all else equal.** Note: These points display the estimated average effect of both treatments. For example, respondents who received the Rohingya treatment are 10.2 percent less likely to donate if they have lived in Dhaka for less than 15 years and 9.9 percent less likely to donate if they have lived in Dhaka for more than 15 years. 95% confidence intervals are also included as horizontal lines.

extensive literature on levels of climate migration and whether climate migrants should be recognized as refugees). We hope this paper will contribute to this growing field given an increased focus on climate migration, especially in the context of migration as a climate adaptation strategy.

Climate change has emerged as an important global public policy issue. However, it is not clear whether climate concerns are equally salient at the domestic level, especially in developing countries that struggle with the challenges of poverty and development. In addressing this question, this paper speaks to the broader issue of why domestic support for some international treaties tends to be spotty. Governments might sign treaties as a way of virtue signaling and ingratiate themselves with important global audiences that have championed these treaties [69–71] but they may not have the local support to implement it. This sort of implementation gap might reflect the fact that international norms are not cohering with local priorities and customs [9, 72]. Worse still, some domestic audiences might view these norms as international and elitist fads that do not address pressing domestic concerns [73]. Indeed, the issue of disconnected elites that are pandering to global audiences figures prominently in the populist discourse [74–76]. While much work pertains to the lack of domestic support to governmental action in response to global policy commitments, this paper extends this argument to the sphere of local support for non-governmental action.

Our paper raises an important question about the lack of political attention to climate issues within developing countries, although many will face severe consequences. In the United States, Canada and Australia, a strong fossil fuel lobby has created a climate countermovement [77]. This sort of industry-inspired backlash to climate issues tends to be missing in many developing countries; although countries such as Malaysia and Indonesia have pushed back against policies to limit deforestation, they have not questioned the science of climate change. Climate change seems to suffer from policy neglect in domestic politics because public attention tends to be focused on either bread and butter issues such as jobs, or cultural issues that often lead to religious or ethnic mobilization. This is worrisome because climate policies and rapid decarbonization will require large-scale mobilization and citizen participation, which could be impeded if citizens view climate change as a "western" issue championed by (elite) individuals and organizations that often depend on foreign funding.

While the Bangladesh government is vocal on climate issues in global forums and has formulated many national-level policies, the salience of climate change in domestic politics remains unclear. Neither the Awami League (the ruling party) or the Bangladesh Nationalist Party (the main opposition party) focuses on climate migrants' issue. This is not limited to Bangladesh only. Its neighboring country India stakes out a climate leadership position in international forums. However, election manifestos of the two major political parties barely contained the mention of climate issues in the recent 2019 elections [78].

We hope our unexpected findings on lower-than-expected local support for climate migrants, even in a climate hotspot country, will spark new research to understand domestic support for climate adaptation. Bangladeshis are very generous despite the level of poverty in Bangladesh. As part of their religious faith, many Muslims regularly provide some sort of zakat or a religious contribution [79]. However, their support for both Rohingyas and climate migrants is below that for generic migrants. This should raise concerns about how Bangladesh will mobilize citizens to address the high level of population displacement that climate change is expected to cause.

This survey experiment has limitations, which highlight areas for further research. First, the high percentages of individuals willing to donate to the charity suggest that the results may suffer from social desirability bias [64]. A future project could compare how individuals say they will give and what they will actually give. Additionally, this study is specific to one slum in

Dhaka. It would be worthwhile to examine whether survey results would differ based on geographic location and proximity to migrant populations. Third, our research design examines whether respondents are willing to donate100 takas. To further validate our study, future work could look at different "price points," especially which are substantially higher than 100 Takas. The reason is that as the financial commitments of the donation increase, respondents might view their support for generic migrant as opposed to the climate migrant and Rohingyas differently.

## Supporting information

**S1 Appendix. Survey questionnaire.**
(DOCX)

**S2 Appendix. OLS regression.**
(DOCX)

**S3 Appendix. Balance table.**
(DOCX)

**S4 Appendix. Ordered probit results.**
(DOCX)

**S5 Appendix. Ordered probit results–full sample.**
(DOCX)

**S6 Appendix. Demographic profile of survey participants.**
(DOCX)

## Author Contributions

**Conceptualization:** Rachel Castellano, Nives Dolšak, Aseem Prakash.

**Data curation:** Rachel Castellano, Nives Dolšak, Aseem Prakash.

**Formal analysis:** Rachel Castellano, Aseem Prakash.

**Funding acquisition:** Nives Dolšak.

**Investigation:** Rachel Castellano, Nives Dolšak.

**Methodology:** Rachel Castellano, Nives Dolšak, Aseem Prakash.

**Project administration:** Rachel Castellano, Nives Dolšak, Aseem Prakash.

**Resources:** Rachel Castellano, Nives Dolšak, Aseem Prakash.

**Software:** Rachel Castellano, Nives Dolšak, Aseem Prakash.

**Supervision:** Nives Dolšak, Aseem Prakash.

**Validation:** Rachel Castellano, Nives Dolšak, Aseem Prakash.

**Visualization:** Rachel Castellano.

**Writing – original draft:** Rachel Castellano, Nives Dolšak, Aseem Prakash.

**Writing – review & editing:** Rachel Castellano, Nives Dolšak, Aseem Prakash.

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
