## [Decision Letter · Decision Letter 0]

16 Dec 2020

PONE-D-20-35549

Willingness to Help Climate Migrants: A Survey Experiment in Bangladesh

PLOS ONE

Dear Dr. prakash,

Thank you for submitting your manuscript to PLOS ONE. After careful consideration, we feel that it has merit but does not fully meet PLOS ONE’s publication criteria as it currently stands. Therefore, we invite you to submit a revised version of the manuscript that addresses the points raised during the review process.

Both reviewers raise serious concerns about the paper. While Reviewer 1 has serious doubts and cannot recommend an invitation to revise, Reviewer 2 is more positive. My own reading is closer to Reviewer 2, which is why I would like to invite you to revise and resubmit the paper. I would like to stress that the changes necessary to make the paper publishable are significant.

In particular, I would like you to

* sharpen the theoretical expectations by engaging more closely with the literature on ‘climate migrants’

* better motivate case selection and discuss issues of external validity

* enhance internal validity by doing appropriate balance tests and address attrition issues

* improve the presentation of findings along the lines suggested by the Reviewers

We look forward to receiving your revised manuscript.

Kind regards,

Bernhard Reinsberg, Ph.D

Academic Editor

PLOS ONE

Journal Requirements:

2) During our internal checks, the in-house editorial staff noted that you conducted research or obtained samples in another country. Please check the relevant national regulations and laws applying to foreign researchers and state whether you obtained the required permits and approvals. Please address this in your ethics statement in both the manuscript and submission information. In addition, please ensure that you have suitably acknowledged the contributions of any local collaborators involved in this work in your authorship list and/or Acknowledgements. Authorship criteria is based on the International Committee of Medical Journal Editors (ICMJE) Uniform Requirements for Manuscripts Submitted to Biomedical Journals - for further information please see here: https://journals.plos.org/plosone/s/authorship.

3) Please include your tables as part of your main manuscript and remove the individual files. Please note that supplementary tables (should remain/ be uploaded) as separate "supporting information" files

4) Please ensure that you refer to Figures 3 & 4 in your text as, if accepted, production will need this reference to link the reader to the figure.

5) We note that you have stated that you will provide repository information for your data at acceptance. Should your manuscript be accepted for publication, we will hold it until you provide the relevant accession numbers or DOIs necessary to access your data. If you wish to make changes to your Data Availability statement, please describe these changes in your cover letter and we will update your Data Availability statement to reflect the information you provide.

Reviewers' comments:

Reviewer's Responses to Questions

**Comments to the Author**

1. Is the manuscript technically sound, and do the data support the conclusions?

Reviewer #1: No

Reviewer #2: Yes

2. Has the statistical analysis been performed appropriately and rigorously? 

Reviewer #1: No

Reviewer #2: Yes

3. Have the authors made all data underlying the findings in their manuscript fully available?

Reviewer #1: No

Reviewer #2: Yes

4. Is the manuscript presented in an intelligible fashion and written in standard English?

Reviewer #1: Yes

Reviewer #2: Yes

5. Review Comments to the Author

Reviewer #1: PONE-D-20-35549

Willingness to Help Climate Migrants: A Survey Experiment in Bangladesh

My general assessment is that the paper contains an interesting idea, which unfortunately has been executed poorly and unconvincingly. In particular, I have some serious concerns regarding the lack of theory and the insufficient execution of the empirical analysis, in particular the survey experiment. This paper needs a lot of work in many areas. I have suggested some but do not feel that a revise and resubmit is appropriate.

Specific comments

Title: I wonder whether the results could be generalized to other slums in Dhaka, let alone to the country as a whole. The title does not even mention that the focus of this study is on a Dhaka slum, and thereby implicitly suggests a broader applicability of the findings at the national level. Hence, given that there is not much in the paper to suggest the findings are more widely applicable, the title should probably be adjusted accordingly.

Abstract: I find the conclusion in the abstract regarding the implications of the findings for support for global policy agendas a little fuzzy and somewhat farfetched, especially, since the main document does not deduce these implications from the empirical evidence.

Introduction

The introduction contains several statements that need better elaboration and/or justification. For instance it is not clear how this paper “…speaks [theoretically] to the broader issue of public support for global policy agendas” (ll. 73-75), when the paper empirically addresses citizens support for a local humanitarian organization aiming at providing health services to ‘climate’ migrants. Furthermore, it is not clear why citizens’ perception in less developed countries of climate change as an elite ‘western’ issue could lead to their lack of support to non-governmental climate action (ll. 83-85).

The authors cite Rigaud et al (2018) to emphasize the effects of climate on future migration flows (ll. 88-90). Given that the focus of the paper is on Bangladesh, the authors could instead mention the Rigaud et al study, which contains a sub- chapter on Bangladesh and estimates that climate induced migration will outpace other internal migrations and predicts that 13.3 million people will be force to move by 2050 under the pessimistic reference scenario. Furthermore, while it is stated that “…[Dhaka] is expected to increase to about 50 million by 2050” (l.95; also a reference is needed here), the Rigaud et al study predicts that climate change will dampen population growth in urban areas such as Dhaka and the river delta south of the city, which will constitute ‘climate’ outmigration hotspots.

The authors state: “Dhaka is the most densely populated city in the world, and the living conditions in Dhaka slums are getting worse as new refugees arrive, about 2,000 a day [The Gardian (15)]” (ll. 95-97). However, it seems that the authors have misquoted the article in the Guardian, since it never mentions ‘climate refuges’, and simply states: “Every day, another 2,000 people move to the Bangladeshi capital. It’s nothing new – for generations Dhaka has been a magnet for those escaping rural poverty – but now climate change is accelerating the race to the city”.

The authors distinguish between three types of migrants, namely ‘religiously persecuted’ ‘climate’, and ‘generic’ (ll. 103-109). While the two first categories could be easily –relatively speaking- defined, the generic one needs to be clarified. That is, what exactly does this category include? Only people, who moved in search for better jobs, economic and educational opportunities as well as standards of living? Or also people, who moved in order to save their lives from political/religious persecution as well as natural disasters? (more on this below)

It is not clear what type of migration the authors study. On the one hand, they talk about climate migrants moving to Dhaka (internal migration) and on the other, refugees from neighbouring Myanmar (Rohingyas) fleeing to Dhaka (international/cross boarder migration) (ll. 103-109). However, in the ‘persecuted migrants treatment’, the Rohingyas are never mentioned and as a result, the respondents were never treated to this specific type of migration (ll. 598-6006). (more on this below)

Migration and Climate Change

The authors do not need to discuss the debate about ‘who counts as a climate migrant (ll.129-141), this debate is well known in the literature. Furthermore, they chose to use the term ‘climate change migrant’ instead of ‘climate refugee’ due to the latter’s contested nature (ll.140-141). Unfortunately, the term “climate migrant” is also problematic since it implies the “pull” of the destination more than the “push” of the source region and carries negative connotations, which reduce the implied responsibility of the international community for their welfare. I believe that the authors should use the term ‘environmental migrant’, which was put forward by IOM in 2007 and is widely used in the relevant literature. That is, “Environmental migrants are persons or groups of persons who, predominantly for reasons of sudden or progressive change in the environment that adversely affects their lives or living conditions, are obliged to leave their habitual homes, or choose to do so, either temporarily or permanently, and who move either within their country or abroad”.

In describing the vulnerability of Bangladesh to climatic changes (ll. 142-150), the authors should use better and more recent references such as the ND-GAIN country Index or the Climate Change Vulnerability Index as well as the Groundswell Report (2018).

The authors state: “Hence, we examine whether Dhaka’s slum dwellers are willing to financially support climate migrants who have joined their community” (ll.158-160). However, a donation to a humanitarian organization, which provides only healthcare services, does not qualify as ‘financial support’ to climate migrants.

Overall, I miss a theoretical argument why “Survey respondents will be more willing to support climate migrants in relation to generic migrants” (H1, LL.183-184), and “Respondents will be less willing to support Rohingyas in relation to generic migrants” (H2, l.204). Especially since the authors state contradictory arguments: on the one hand, respondents might be will to support climate migrants due to empathy (ll.161-165), and on the other, they might not be willing to support them due to competition over scarce public services such as health (ll. 172-174). The authors hence need to develop the arguments more thoroughly, thereby directing the reader to the hypotheses that are tested afterwards. In addition, the survey experiment is not appropriate for testing H2 since it did not explicitly mention the Rohingyas, but rather people who were displaced by religious violence!

Methodology

Selection of the survey site: The survey took place in the Korail slum in Dhaka. Given that Dhaka city has more than 3,300 slums (inhabited by an estimated 6.5 million people), a justification of the selection of this site is needed in order for the findings to be generalized to other slums and to allow for the conclusions the authors reach regarding the lack of support for global policy agendas. In other words, would the authors obtain similar results if they conducted their survey experiment in, say, the Sattola slum instead? Both Korail and Sattola are established slums in the North Dhaka City Corporation, situated beside elite, residential areas of Dhaka city, where land prices are high and there is high potential for urban development. However, in Korail slum, about 47% slum dwellers obtain medical service from NGOs, which is much higher compared to the Sattola slum dwellers where only 33% get medical services from NGOs. In addition, while many Korail dwellers have been relocated there due to major eviction drives in other Dhaka slums (Korail has never fallen victim to evictions owing to its strong political backing), in Sattola most migrants come from different disaster-prone and river-eroded areas such as the northern chars and the coastal belt.

Selection of respondents: the reader should know more about the procedure and criteria how respondents were selected. E.g. did the data collection cover the entire Korail slum or only specific neighbourhoods (which ones)? was cluster sampling used? The authors, hence, need to describe better the survey design. I know how challenging it is to reach and interview people especially in challenging locations, such as Bangladesh. I do not mean to imply that one needs to have the same level of sophistication in sampling procedures as we would expect from survey research in OECD countries. However, it is important to communicate just how the sample was constructed, beyond and above, showing that it is balanced in sex and closely matches national averages in religion and occupation.

Frame experiment (Appendix S1)

Generic Frame: the ‘generic frame’ includes ‘people, who for a variety of reasons are forced to leave their homes and move to other areas in search for a better future or a safer place’ (ll. 589-591). However, this category of migrants presents severe identification problems for the frame experiment, since it is not mutually exclusive from the other two categories of migrants. Take for instance the ‘climate’ migrant frame: since, it is extremely difficult to disentangle the environmental from the economic reason of migration, it is possible that for many respondents the ‘generic’ frame contains also climate induced migration. Furthermore, and perhaps more importantly, the ‘generic’ frame contains information, i.e., “They are in urgent need of humanitarian assistance.” (l. 592), which is missing in the two treatment frames. This information makes the ‘generic’ frame much stronger than the two treatment frames, affecting hence the responses and the empirical results.

‘Persecuted Migrants Treatment’: given that ‘persecuted migrants treatment’ never mentioned the Rohingyas, it is very likely that the respondents perceived the ‘religious violence’ to apply to the domestic religious/ethnic minorities. In Bangladesh, even though the government publicly supports(ed) freedom of religion, still Hindu, Christian, and Buddhist minorities experience(d) discrimination and sometimes violence from the Muslim majority. Hence, it is not surprising that the survey respondents, who are mostly Muslims (98%), would not support giving to a charity that aims at helping these minority groups.

‘Climate Migrants Treatment’: the authors state that heavy flooding is associated with climate change (ll. 610). Why? Bangladeshis know that the monsoon season always brings heavy rainfall to the country, which is critical for replenishing water supplies and helps farmers, but it can also cause great damage! Furthermore, the mentioning of climate change might lead to biased responses, as rightly the authors point out (ll. 293-297).

Results

The authors need to discuss the possibility that the ‘high willingness to donate’ observed in their data ((ll.274-282)) might be due to social desirability bias.

The interpretation of the finding of low support for climate migrants is quite superficial (ll.287-292).

There are also some contradictions. For instance, they authors attribute the low probability of giving to the ‘persecuted migrants’ (i.e., Rohingyas) to the media and the Bangladesh government, which have perpetuated harmful narratives about these migrants (ll. 301-305), even though they do not find any support for the conditioning effect of media consumption! (ll. 312-317).

Conclusion

A large part of this section is quite unrelated to the research question of this paper and confusing (ll. 347-379).

Reviewer #2: This manuscript sets out to investigate the preferences of slum dwellers in Dhaka, Bangladesh regarding three categories of migrants: generic migrants, climate migrants, and religiously-persecuted refugees. Specifically, the researchers embedded a survey experiment in a survey of 1,800 respondents. Respondents were provided vignettes about a fictitious humanitarian group seeking to raise funds for migrants who were randomly described as belonging to one of the three categories above. Respondents were then asked whether they were willing to donate funds to support the charity. The authors find high levels of support for willingness to contribute funds when the migrants in question were generic migrants: 86 percent of this treatment arm agreed to support the humanitarian charity. Against expectations, support plummeted for climate migrants (61 percent) as well as for religiously persecuted migrants (77 percent).

Overall, this is a nicely conceptualized and executed study, and it contributes to scholarly knowledge in an important domain where empirical research is scarce. Nevertheless, I would urge the authors to address the following questions and suggestions:

1. The manuscript could do more to theoretically motivate its predictions regarding climate migrants (H1) and its interpretations of the empirical tests of H1. The authors argue that poor Dhaka residents should theoretically be more in favor of climate migrants since the Bangladeshi government has prioritized climate change and has highlighted the plight of climate refugees in the past. That government actions shape citizen preferences is a plausible conjecture. At the same time, the literature on migration (both cross-border and internal) has clearly identified economic competition to be an important predictor of nativist preferences toward migrants. Both job market competition and fiscal pressures (e.g., competition for public housing, education, employment, etc.) can lead locals to oppose the entry of migrants. Climate migrants are likely a special category of migrants, since presumably they are permanent migrants who have little ability to return to their “homes.” Hence, they might produce pronounced economic threats to locals. By contrast, it is quite possible that generic migrants are conceptualized as temporary or seasonal migrants (or at the very least not as permanently dependent on the welfare state in Dhaka as climate migrants). In this theoretical light, the pronounced opposition to climate migrants (compared to the generic migrants) that the manuscript documents might appear to be quite reasonable and rational. These are the types of migrants who plausibly pose the starkest economic threats to the poor slum-dwellers in the study’s sample. I encourage the authors to discuss and probe this possibility, both theoretically and in their analysis of the results. One idea would be to analyze whether the treatment effects vary by respondents’ household monthly incomes.

2. The manuscript should do more to interpret the “willingness to donate” outcome measure that is used in all of the primary analyses. Since this is a self-reported measure that does not have a behavioral component (i.e., an observed measure of how much subjects would have actually donated if given the option), it is a bit unclear how readers should interpret this measure. Clearly, the baseline levels of professed support are very high. Given the low socio-economic status of the sample, it is unlikely that such a high proportion of respondents (75 percent across all treatment arms) would in reality donate funds to charities. Are the authors concerned about survey response bias? Of particular concern is the possibility that survey response bias is lower for religiously persecuted migrants and climate migrants, which may explain why subjects are more willing to deny donating funds to support these particular types of migrants. The authors could comment on these possibilities and ideally offer some kinds of empirical evidence to explore and rule them out. More broadly, the manuscript would be stronger if it provided guidance to readers on how to interpret the self-reported willingness to pay measure.

3. The manuscript repeatedly describes the frame regarding religiously persecuted migrants as the Rohingya frame. In Appendix S1, however, the “Persecuted Migrants Treatment” does not appear to specifically mention that these migrants are Rohingya migrants. Of course, in the Bangladesh context the Rohingya are indeed the main type of religiously persecuted migrants. But if the treatment frame did not use the term Rohingya, it is not immediately clear that respondents would have assumed that the migrants in the survey vignette were Rohingya. It is fine if the authors want to retain the Rohingya terminology, but they should make clear to readers their rationale for doing so and explain why respondents would likely not have considered any other types of religiously persecuted migrants in this context.

4. Because the manuscript limits its analysis to only those respondents (1,443 of the 1,800) who correctly answered three comprehension questions, it ends up dropping subjects in what appears to be an unbalanced manner (see p. 12). Appendix S3 presents summary statistics of key variables but does not present formal statistical tests of balance. I would recommend presenting formal tests of balance. If unbalanced, it may make sense to present the results in the appendix of all of the primary analyses utilizing the full sample in the study.

6. PLOS authors have the option to publish the peer review history of their article (what does this mean?). If published, this will include your full peer review and any attached files.

Reviewer #1: No

Reviewer #2: No

---

## [Author Response · Author response to Decision Letter 0]

29 Jan 2021

Rebuttal Memo 

Willingness to help climate migrants: 

A survey experiment in the Korail Slum of Dhaka, Bangladesh

PONE-D-20-35549

We thank the reviewers for their exceptionally detailed, thoughtful, and constructive feedback. We are enclosing a rebuttal memo detailing the reviewers’ suggestions and outlining specific ways we address them in the revised manuscript.

Sincerely,

Authors

Reviewer 1

1. Title: I wonder whether the results could be generalized to other slums in 

Dhaka, let alone to the country as a whole. The title does not even mention that the focus of this study is on a Dhaka slum, and thereby implicitly suggests a broader applicability of the findings at the national level. Hence, given that there is not much in the paper to suggest the findings are more widely applicable, the title should probably be adjusted accordingly.

Response:

Thank you. The new title is: “Willingness to help climate migrants: A survey experiment in the Korail Slum of Dhaka, Bangladesh”.

2. Abstract: I find the conclusion in the abstract regarding the implications of the 

findings for support for global policy agendas a little fuzzy and somewhat farfetched, especially, since the main document does not deduce these implications from the empirical evidence.

Response:

Fair point. We have removed this sentence from the abstract.

3. Introduction: The introduction contains several statements that need better 

elaboration and/or justification. For instance it is not clear how this paper “…speaks [theoretically] to the broader issue of public support for global policy agendas” (ll. 73-75), when the paper empirically addresses citizens support for a local humanitarian organization aiming at providing health services to ‘climate’ migrants. Furthermore, it is not clear why citizens’ perception in less developed countries of climate change as an elite ‘western’ issue could lead to their lack of support to non-governmental climate action (ll. 83-85).

Response:

Thanks for raising these issues, which we have clarified in the revised paper. There are two issues: why should citizens oppose global agendas, and second why should they oppose NGOs that are involved in climate action.

Our paper speaks to a broader debate on citizen perceptions of salient global issues, and how they form opinion about actors, both governmental and nongovernmental, that work domestically on these issues. Why should this matter? International Relations scholars have examined whether international treaties require domestic support (for example, see the literature on “two-level” games as well as “audience costs”). The reason is that international treaties obligate governments to enact and enforce policies domestically. Governments fear high political costs when citizens believe that new policies militate against their interests and beliefs. Citizens might entertain the perception that their government signed on to these treaties because it succumbed to international (often viewed as, Western or elite) pressure. Recent examples of citizen opposition include the issues of gender equality, same-sex marriage, the migration crisis in Europe, and Brexit. In some countries, international trade agreements are also viewed as elite impositions that enrich global corporations at the expense of workers. Broadly, the populist rhetoric against globalization falls in this category. Climate change is an important global issue but policies such as carbon taxes have (unfortunately) invited populist backlash even in developed countries (see the “yellow vest” protests or the defeat of two carbon tax initiatives in the state of Washington). The issue of climate migration is even more complex, given the political opposition to migration in many countries.

Why should citizens not support non-governmental organizations (NGOs) that work on humanitarian issues, irrespective of whether their action is motivated by a global policy concern? The literature suggests that citizens may sometimes think of NGOs as the part of the establishment, and not as local organization that help local communities. There is an emerging literature in development studies on “democracy recession.” In the last two decades, there has been a massive crackdown against NGOs across countries. Governments have incentives to crackdown when they perceive NGOs are working with their political opponents. They feel emboldened to crack down when they perceive that NGOs do not have citizen support (in other words, the political costs of cracking down are low). 

Why the lack of support for NGOs? Sometimes citizens believe that NGOs work for western agendas instead of local concerns. Scholars term this as the “NGOization” of the civil society. In the 1990, as foreign donors began routing aid through NGOs as opposed to local governments, NGOs became visible in public service delivery –sometimes even more than the local government. For example, NGOs flooded Haiti after the 2010 earthquake. Not surprisingly, Haiti has acquired the label of the “Republic of NGOs.” Competition among NGOs for funding meant that NGOs were perceived as working on agendas dictated by their western donors. And there are also cases of NGO misconduct such as the recent Oxfam scandal. The lavish lifestyle of some NGOs also contributed to the perception among some citizens that NGOs are elites. Thus, citizens sometimes become wary of even local humanitarian NGOs especially when they work on “global” agendas.

Nevertheless, it is not clear whether Bangladesh citizens will support NGOs providing humanitarian services to climate migrants. After all, Bangladesh is directly impacted by climate change. Further, unlike Haiti where the citizen anger is often directed at foreign (and rich) NGOs, the hypothetical NGO in our survey experiment was dependent on local resources (as opposed to foreign funding). Hence, our survey findings are contrary to our theoretical expectations, as we note in the paper. We suggest that perhaps this is because citizens have already formed their opinion about climate change and NGOs working on this subject. In particular, given the media publicity on climate issues and the constant refrain about its global dimensions, citizens probably feel that it is an elite issue or that NGOs in fact have foreign funding. Because we do not specifically explore the reasons for distrust in NGOs, we hope future work will explore this issue in greater detail.

We have incorporated the above discussion both in the introduction and conclusion. In addition, we have included new citations to support our argument:

Bob, C. (2005). The Marketing of Rebellion: Insurgents, Media and Transnational Support. Cambridge University Press.

Dupuy, K., Ron, J. & Prakash, A. (2015). Hands off my regime! Governments’ restrictions on foreign aid to non-governmental organizations in poor and middle-income countries. World Development, 84: 299-311. 

Edwards, M. & Hulme, D. (1996). Too close for comfort? The impact of official aid on non-governmental organizations. World Development, 24: 961-973. 

Hearn, J. (1998). The ‘NGO‐isation’ of Kenyan society: USAID & the restructuring of health care. Review of African Political Economy, 25(75): 89-100.

Kristoff, M., & Panarelli, L. 2010. Haiti: A Republic of NGOs? Peace Brief 23. Washington, DC: United States Institute of Peace.

4. The authors cite Rigaud et al (2018) to emphasize the effects of climate on future 

migration flows (ll. 88-90). Given that the focus of the paper is on Bangladesh, the authors could instead mention the Rigaud et al study, which contains a sub- chapter on Bangladesh and estimates that climate induced migration will outpace other internal migrations and predicts that 13.3 million people will be force to move by 2050 under the pessimistic reference scenario. Furthermore, while it is stated that “…[Dhaka] is expected to increase to about 50 million by 2050” (l.95; also a reference is needed here), the Rigaud et al study predicts that climate change will dampen population growth in urban areas such as Dhaka and the river delta south of the city, which will constitute ‘climate’ out-migration hotspots.

Response: 

Thank you. We have included a citation of Raigaud et al. to note that climate induced migration will outpace other internal migrations. 

5. The authors state: “Dhaka is the most densely populated city in the world, and 

the living conditions in Dhaka slums are getting worse as new refugees arrive, about 2,000 a day [The Gardian (15)]” (ll. 95-97). However, it seems that the authors have misquoted the article in the Guardian, since it never mentions ‘climate refuges’, and simply states: “Every day, another 2,000 people move to the Bangladeshi capital. It’s nothing new – for generations Dhaka has been a magnet for those escaping rural poverty – but now climate change is accelerating the race to the city”.

Response: 

Thank you; we have corrected the text. 

6. The authors distinguish between three types of migrants, namely ‘religiously 

persecuted’ ‘climate’, and ‘generic’ (ll. 103-109). While the two first categories could be easily –relatively speaking- defined, the generic one needs to be clarified. That is, what exactly does this category include? Only people, who moved in search for better jobs, economic and educational opportunities as well as standards of living? Or also people, who moved in order to save their lives from political/religious persecution as well as natural disasters? (more on this below)

Response: 

Great point – we have expanded the discussion on generic migration, including those who migrate in search of any factor including better economic and educational opportunities. The objective of the generic category is to provide a benchmark (or reference category) to assess if the willingness to support the charity changes when a specific migration driver is identified in the treatment frame. We have noted this point in the revised manuscript. 

7. It is not clear what type of migration the authors study. On the one hand, they 

talk about climate migrants moving to Dhaka (internal migration) and on the other, refugees from neighbouring Myanmar (Rohingyas) fleeing to Dhaka (international/cross boarder migration) (ll. 103-109). However, in the ‘persecuted migrants treatment’, the Rohingyas are never mentioned and as a result, the respondents were never treated to this specific type of migration (ll. 598-6006). (more on this below)

Response: 

In consultation with local survey firm, we employed the phrase “persecuted minority” instead of Rohingyas in the survey instrument. The local firm felt that the phrase Rohingya is extremely volatile in Bangladesh. Indeed, the government is seeking to relocate Rohingyas on an island which is prone to storms. Thus, while the persecuted minority clearly signals that we are asking about Rohingya, it will not unleash the passion of the respondent. 

We recognize that Buddhists and Hindus could also be considered to be persecuted minorities (although less under the current Awami League regime of Sheikh Hasina) and we raised the issue with the survey firm. We were advised that Hindus and Buddhist tend not to migrate to Dhaka but instead head to say India. We have included this brief discussion in the revised paper.

8. Migration and Climate Change:

The authors do not need to discuss the debate about ‘who counts as a climate migrant (ll.129-141), this debate is well known in the literature. Furthermore, they chose to use the term ‘climate change migrant’ instead of ‘climate refugee’ due to the latter’s contested nature (ll.140-141). Unfortunately, the term “climate migrant” is also problematic since it implies the “pull” of the destination more than the “push” of the source region and carries negative connotations, which reduce the implied responsibility of the international community for their welfare. I believe that the authors should use the term ‘environmental migrant’, which was put forward by IOM in 2007 and is widely used in the relevant literature. That is, “Environmental migrants are persons or groups of persons who, predominantly for reasons of sudden or progressive change in the environment that adversely affects their lives or living conditions, are obliged to leave their habitual homes, or choose to do so, either temporarily or permanently, and who move either within their country or abroad”.

Response: 

In the revised paper, we have noted the debate on climate refugees and climate migration more generally. Since the paper is focused on climate change, we believe that the use of the phrase climate migrant is appropriate. We have, however, incorporated your point about IOM, suggesting the use of the term environmental migrant.

9. In describing the vulnerability of Bangladesh to climatic changes (ll. 142-150), 

the authors should use better and more recent references such as the ND-GAIN country Index or the Climate Change Vulnerability Index as well as the Groundswell Report authored by Raigud et al. (2018).

Response: 

Thank you! We have referenced these indexes in the revised version. 

10. The authors state: “Hence, we examine whether Dhaka’s slum dwellers are 

willing to financially support climate migrants who have joined their community” (ll.158-160). However, a donation to a humanitarian organization, which provides only healthcare services, does not qualify as ‘financial support’ to climate migrants.

Response: 

Fair point. We have modified the language to make clear that the support is for healthcare services.

10. Overall, I miss a theoretical argument why “Survey respondents will be more 

willing to support climate migrants in relation to generic migrants” (H1, LL.183-184), and “Respondents will be less willing to support Rohingyas in relation to generic migrants” (H2, l.204). Especially since the authors state contradictory arguments: on the one hand, respondents might be will to support climate migrants due to empathy (ll.161-165), and on the other, they might not be willing to support them due to competition over scarce public services such as health (ll. 172-174). The authors hence need to develop the arguments more thoroughly, thereby directing the reader to the hypotheses that are tested afterwards. In addition, the survey experiment is not appropriate for testing H2 since it did not explicitly mention the Rohingyas, but rather people who were displaced by religious violence!

Response: 

We offer several different perspectives on why respondents might or might not be willing to support an organization that supports migrants, such as empathy-driven giving and competition over scarce services. Thus, we do not have a theoretical position on which perspective will prevail; this eventually needs to be resolved empirically. We clarify these points as they relate to our hypotheses. Regarding Rohingyas, see please the response to point 7.

11. Methodology

Selection of the survey site: The survey took place in the Korail slum in Dhaka. Given that Dhaka city has more than 3,300 slums (inhabited by an estimated 6.5 million people), a justification of the selection of this site is needed in order for the findings to be generalized to other slums and to allow for the conclusions the authors reach regarding the lack of support for global policy agendas. In other words, would the authors obtain similar results if they conducted their survey experiment in, say, the Sattola slum instead? Both Korail and Sattola are established slums in the North Dhaka City Corporation, situated beside elite, residential areas of Dhaka city, where land prices are high and there is high potential for urban development. However, in Korail slum, about 47% slum dwellers obtain medical service from NGOs, which is much higher compared to the Sattola slum dwellers where only 33% get medical services from NGOs. In addition, while many Korail dwellers have been relocated there due to major eviction drives in other Dhaka slums (Korail has never fallen victim to evictions owing to its strong political backing), in Sattola most migrants come from different disaster-prone and river-eroded areas such as the northern chars and the coastal belt.

Response: 

Thank you for bringing up the issue of generalizability. We expanded the discussion in the “Methodology” section that addresses the case selection of the Korail slum. We also comment on this limitation in our conclusion and discuss the need for further research in other parts of Dhaka and Bangladesh. 

12. Selection of respondents: the reader should know more about the procedure and 

criteria how respondents were selected. E.g. did the data collection cover the entire Korail slum or only specific neighbourhoods (which ones)? was cluster sampling used? The authors, hence, need to describe better the survey design. I know how challenging it is to reach and interview people especially in challenging locations, such as Bangladesh. I do not mean to imply that one needs to have the same level of sophistication in sampling procedures as we would expect from survey research in OECD countries. However, it is important to communicate just how the sample was constructed, beyond and above, showing that it is balanced in sex and closely matches national averages in religion and occupation.

Response:

We have included the following information in the revised manuscript. “The survey team collected data from almost entire Korail slum. They started with identifying five blocks based on the scoping study. The entire slum was grouped into 20 clusters based on these blocks. Employing a single-stage cluster sampling considering gender, religion and occupations, the team interviewed 100 respondents in each cluster. When respondents did not give their consent to take part in the interview, the survey team moved to another respondent. Only one household member in each family was interviewed in this study.”

13. Frame experiment (Appendix S1)

Generic Frame: the ‘generic frame’ includes ‘people, who for a variety of reasons are forced to leave their homes and move to other areas in search for a better future or a safer place’ (ll. 589-591). However, this category of migrants presents severe identification problems for the frame experiment, since it is not mutually exclusive from the other two categories of migrants. Take for instance the ‘climate’ migrant frame: since, it is extremely difficult to disentangle the environmental from the economic reason of migration, it is possible that for many respondents the ‘generic’ frame contains also climate induced migration. 

Response: 

Our objective is to assess if the support for the NGO changes if specific drivers of migration are identified. Frames are identical, except for one factor—the information about the migration driver. Therefore, they are not supposed to be mutually exclusive. In the generic frame we do not provide information of any specific migration driver: it merely assesses public support for an NGO that is providing health services to any migrant which would serve as the reference category or benchmark to assess other frames. But if the migration driver does matter (because it generates empathy or fear) in generating public support, then it has important policy implications. For example, if migration is caused by climate change, then the governments will need to sensitize the public that migrants are “victims” of exogenous factors which are beyond their control. We have briefly elaborated on this issue in the revised paper.

14. Furthermore, and perhaps more importantly, the ‘generic’ frame contains 

information, i.e., “They are in urgent need of humanitarian assistance.” (l. 592), which is missing in the two treatment frames. This information makes the ‘generic’ frame much stronger than the two treatment frames, affecting hence the responses and the empirical results.

Response:

We cross-checked the survey and confirm that following sentence is identical in the generic frame and the two treatment frames: “There is a very large number of poor people in urgent need of humanitarian assistance.” Hence, generic frame is not stronger than the two treatment frames.

15. ‘Persecuted Migrants Treatment’: given that ‘persecuted migrants treatment’ 

never mentioned the Rohingyas, it is very likely that the respondents perceived the ‘religious violence’ to apply to the domestic religious/ethnic minorities. In Bangladesh, even though the government publicly supports(ed) freedom of religion, still Hindu, Christian, and Buddhist minorities experience(d) discrimination and sometimes violence from the Muslim majority. Hence, it is not surprising that the survey respondents, who are mostly Muslims (98%), would not support giving to a charity that aims at helping these minority groups.

Response: 

Thank you, this is an important point! After much discussion with our research partners in Dhaka, we chose not to include the term Rohingya because of the emotional salience the term currently carries in Bangladesh. Indeed, the level of hostility towards Rohingya, encouraged by the government, is quite high. The local survey firm felt that Rohingya could become a trigger word and provoke an extreme negatively emotional reaction. Hence, they advised that we could still talk about Rohingyas in terms of “religious persecuted” minority without mentioning them directly. Thus, arguably, if we had employed the phrase Rohingya as opposed to a religious persecuted minority, the decline in support in this frame (in relation to the generic frame) might have been even higher. Thus, we consider our results for this frame to be conservative.

We recognize that several minorities in Bangladesh have experienced religious persecution. Yet, there are no media reports of large-scale violence against Hindus and Buddhists under the current Awami League regime. Indeed, this regime has cracked down on Islamic fundamentalist group that worked with Pakistani Army during the Liberation war and were often in the forefront of fomenting violence against minorities. 

Thus, to guard against any confusion on the nature of religiously persecuted minorities, we chose the language in the treatment carefully: “religious violence is causing a large displacement of people.” We are confident that respondents recognized this as Rohingya.

As a practical matter, the out-migration of Hindus has tended to flow towards India (where it has become a political issue in the context of the Citizenship Bill) and is therefore not a major political issue within Bangladesh. We have elaborated on this issue in our “Introduction” as well as the Methodology section.

15. ‘Climate Migrants Treatment’: the authors state that heavy flooding is 

associated with climate change (ll. 610). Why? Bangladeshis know that the monsoon season always brings heavy rainfall to the country, which is critical for replenishing water supplies and helps farmers, but it can also cause great damage! Furthermore, the mentioning of climate change might lead to biased responses, as rightly the authors point out (ll. 293-297).

Response:

Fair point. As scholars note, climate change is likely to accentuate the frequency and severity of even regularly occurring weather events such as the annual flooding. We have noted this in the revised paper.

16. Results:

The authors need to discuss the possibility that the ‘high willingness to donate’ observed in their data ((ll.274-282)) might be due to social desirability bias.

Response: 

Fair point. We have noted this in the paper. 

17. The interpretation of the finding of low support for climate migrants is quite 

superficial (ll.287-292). There are also some contradictions. For instance, they authors attribute the low probability of giving to the ‘persecuted migrants’ (i.e., Rohingyas) to the media and the Bangladesh government, which have perpetuated harmful narratives about these migrants (ll. 301-305), even though they do not find any support for the conditioning effect of media consumption! (ll. 312-317).

Response: 

If media is saturated with negative stories, then media will not have a conditioning effect as along as people have a baseline exposure. We have clarified this point in the revised paper.

17. Conclusion

A large part of this section is quite unrelated to the research question of this paper and confusing (ll. 347-379).

Response: 

Thank you for your feedback. We reworked the conclusion and expanded upon many of the points and limitations that you commented on. 

 

Reviewer #2

1. This manuscript sets out to investigate the preferences of slum dwellers in 

Dhaka, Bangladesh regarding three categories of migrants: generic migrants, climate migrants, and religiously-persecuted refugees. Specifically, the researchers embedded a survey experiment in a survey of 1,800 respondents. Respondents were provided vignettes about a fictitious humanitarian group seeking to raise funds for migrants who were randomly described as belonging to one of the three categories above. Respondents were then asked whether they were willing to donate funds to support the charity. The authors find high levels of support for willingness to contribute funds when the migrants in question were generic migrants: 86 percent of this treatment arm agreed to support the humanitarian charity. Against expectations, support plummeted for climate migrants (61 percent) as well as for religiously persecuted migrants (77 percent).

Overall, this is a nicely conceptualized and executed study, and it contributes to scholarly knowledge in an important domain where empirical research is scarce. Nevertheless, I would urge the authors to address the following questions and suggestions:

Response: 

Thank you! 

2. The manuscript could do more to theoretically motivate its predictions 

regarding climate migrants (H1) and its interpretations of the empirical tests of H1. The authors argue that poor Dhaka residents should theoretically be more in favor of climate migrants since the Bangladeshi government has prioritized climate change and has highlighted the plight of climate refugees in the past. That government actions shape citizen preferences is a plausible conjecture. At the same time, the literature on migration (both cross-border and internal) has clearly identified economic competition to be an important predictor of nativist preferences toward migrants. Both job market competition and fiscal pressures (e.g., competition for public housing, education, employment, etc.) can lead locals to oppose the entry of migrants. Climate migrants are likely a special category of migrants, since presumably they are permanent migrants who have little ability to return to their “homes.” Hence, they might produce pronounced economic threats to locals. By contrast, it is quite possible that generic migrants are conceptualized as temporary or seasonal migrants (or at the very least not as permanently dependent on the welfare state in Dhaka as climate migrants). In this theoretical light, the pronounced opposition to climate migrants (compared to the generic migrants) that the manuscript documents might appear to be quite reasonable and rational. These are the types of migrants who plausibly pose the starkest economic threats to the poor slum-dwellers in the study’s sample. I encourage the authors to discuss and probe this possibility, both theoretically and in their analysis of the results. One idea would be to analyze whether the treatment effects vary by respondents’ household monthly incomes.

Response: 

Thank you for this insightful comment and for highlighting a plausible explanation of our results. We agree that climate migrants may pose the largest economic threat because of perceived permanence. We have included this point in the revised paper.

3. The manuscript should do more to interpret the “willingness to donate” outcome 

measure that is used in all of the primary analyses. Since this is a self-reported measure that does not have a behavioral component (i.e., an observed measure of how much subjects would have actually donated if given the option), it is a bit unclear how readers should interpret this measure. Clearly, the baseline levels of professed support are very high. Given the low socio-economic status of the sample, it is unlikely that such a high proportion of respondents (75 percent across all treatment arms) would in reality donate funds to charities. Are the authors concerned about survey response bias? Of particular concern is the possibility that survey response bias is lower for religiously persecuted migrants and climate migrants, which may explain why subjects are more willing to deny donating funds to support these particular types of migrants. The authors could comment on these possibilities and ideally offer some kinds of empirical evidence to explore and rule them out. More broadly, the manuscript would be stronger if it provided guidance to readers on how to interpret the self-reported willingness to pay measure.

Response: 

Thank you for bringing this up. We elaborate on our discussion of potential social desirability bias in the revised paper.

4. The manuscript repeatedly describes the frame regarding religiously persecuted 

migrants as the Rohingya frame. In Appendix S1, however, the “Persecuted Migrants Treatment” does not appear to specifically mention that these migrants are Rohingya migrants. Of course, in the Bangladesh context the Rohingya are indeed the main type of religiously persecuted migrants. But if the treatment frame did not use the term Rohingya, it is not immediately clear that respondents would have assumed that the migrants in the survey vignette were Rohingya. It is fine if the authors want to retain the Rohingya terminology, but they should make clear to readers their rationale for doing so and explain why respondents would likely not have considered any other types of religiously persecuted migrants in this context.

Response: 

Thank you for bringing this up. We discuss our rationale for not using the term ‘Rohingya’ in the “Methodology” section. 

5. Because the manuscript limits its analysis to only those respondents (1,443 of 

the 1,800) who correctly answered three comprehension questions, it ends up dropping subjects in what appears to be an unbalanced manner (see p. 12). Appendix S3 presents summary statistics of key variables but does not present formal statistical tests of balance. I would recommend presenting formal tests of balance. If unbalanced, it may make sense to present the results in the appendix of all of the primary analyses utilizing the full sample in the study.

Response: 

Thank you for bringing this up. Our results hold when utilizing the full sample in the study. We have included both results in the revised manuscript.

---

## [Decision Letter · Decision Letter 1]

17 Feb 2021

PONE-D-20-35549R1

Willingness to help climate migrants:  A survey experiment in the Korail Slum of Dhaka, Bangladesh

PLOS ONE

Dear Dr. prakash,

Thank you for submitting your manuscript to PLOS ONE. After careful consideration, we feel that it has merit but does not fully meet PLOS ONE’s publication criteria as it currently stands. Therefore, we invite you to submit a revised version of the manuscript that addresses the points raised during the review process.

As you notice, the reviewers are more positive about your manuscript following your revisions. I agree with them but insist that you resubmit a revised paper addressing their remaining comments and concerns. Especially Reviewer 2 has some important questions that need to be addressed. Specifically, you should back up your assumptions about how individuals learn about policy positions of their government regarding climate migrants. Furthermore, you should check your survey for cues from respondents in the control group about what they think when thinking about migrants in general. This would help further increase the validity of your study.

We look forward to receiving your revised manuscript.

Kind regards,

Bernhard Reinsberg, Ph.D

Academic Editor

PLOS ONE

Reviewers' comments:

Reviewer's Responses to Questions

**Comments to the Author**

1. If the authors have adequately addressed your comments raised in a previous round of review and you feel that this manuscript is now acceptable for publication, you may indicate that here to bypass the “Comments to the Author” section, enter your conflict of interest statement in the “Confidential to Editor” section, and submit your "Accept" recommendation.

Reviewer #2: (No Response)

Reviewer #3: (No Response)

2. Is the manuscript technically sound, and do the data support the conclusions?

Reviewer #2: Yes

Reviewer #3: Partly

3. Has the statistical analysis been performed appropriately and rigorously? 

Reviewer #2: Yes

Reviewer #3: Yes

4. Have the authors made all data underlying the findings in their manuscript fully available?

Reviewer #2: Yes

Reviewer #3: Yes

5. Is the manuscript presented in an intelligible fashion and written in standard English?

Reviewer #2: Yes

Reviewer #3: Yes

6. Review Comments to the Author

Reviewer #2: This is a strong revision and I would be delighted to see this research published, pending a few minor revisions listed below. I paid particular attention to the author(s)’ responses to the prior referee reports, and to their alterations of the manuscript to reflect their adjustments to critiques. In particular, the theoretical framework in the revised manuscript is considerably strengthened, the link between theory and empirics is more focused, and the empirical evidence remains robust to the different sensitivity checks that the author(s) have now introduced. Overall, I am impressed by the constructive revisions undertaken by the author(s) and see the author(s) as admirably responsive to the prior concerns raised in review.

Minor revisions:

1. In l.106 “especially if it involves citizens to incur private costs” would read better as “especially if it involves citizens incurring private costs.”

2. In response to R1’s excellent comments, the author(s) now address the issue of religious similarities and differences between the migrants and respondents quite extensively in the manuscript, noting that the Rohingya share the same Islamic faith as the majority of Bangladeshis (e.g., p.10), that respondents would likely not have associated the migrants with religious minorities like Hindus (p.12), and that respondents could have been expected to be “sympathetic to their co-religionists” (p.14, l.304). In neighboring India, which is frequently referenced in the manuscript, existing public opinion work shows that city residents belonging to the majority religious group (in the Indian case, Hindus) do not discriminate again rural-to-urban migrants based on religious profiles. Instead, residents from the dominant religious group appear to care mainly about the economic impact of migration. This is in fact consistent with what the author(s) find in the Bangladesh context: slum residents do not appear to heed religious concerns in responding to migration. The manuscript would be strengthened by citing and referencing these points as it would link the manuscript to broader debates in comparative politics.

Citation: Gaikwad, N. and Nellis, G. (2017), “The Majority‐Minority Divide in Attitudes toward Internal Migration: Evidence from Mumbai.” American Journal of Political Science, 61: 456-472.

3. In l.343, the sentence is missing a closed parenthesis mark.

Overall, I continue to believe that this manuscript tackles a very important subject, and I expect it to foreshadow new work on climate change and migration.

Reviewer #3: Review of:

Willingness to Help Climate Migrants: A Survey Experiment in Bangladesh

This paper examines individuals’ willingness to provide assistance to climate migrants as compared to other types of migrants by means of a survey-embedded experiment conducted in a slum in Dhaka. The paper is well-written and addresses an important and very timely question. I believe that this study can make a contribution to the literature after addressing the issues below.

Theory

I think the authors need to provide a more specific discussion of the mechanism(s) underlying their main hypothesis, especially “H1: Survey respondents will be more willing to support climate migrants in relation to generic migrants.” The arguments that the authors provide (line 222: “extensive focus on climate change in media and the strong advocacy by the Bangladesh government in global forums”) are based on the assumption that citizens of Korail slum take their cues from the media and policymakers and will formulate their willingness to support (climate) migrants accordingly. However, this not only requires that slum residents have access to these specific information, but are also well-informed about what their national politicians advocate in global policy arenas. I think the authors could expend more efforts to provide more micro-level arguments as to why they think climate migrants might be perceived in more favorable lights than other (generic) migrant types.

Methodology

My main issue with this manuscript is the design of the generic frame. The problem is that the lack of a specific driver of migration in this frame introduces the possibility for survey respondents to think about all sorts of migrant type, including climate migrants as well as both domestic and international migrants. I think this heterogeneity, but more importantly, the impossibility to identify what type of migrant the respondent is thinking about when answering whether they’d like to help that group of migrants, makes it very hard to interpret the outcome variable itself. What is the level of support we are benchmarking support for climate migrants against? We know from the literature that different types of migrants are seen in very different ways (as this manuscript also argues). Therefore, it is important to know how individuals perceive climate migrants compared to other (specific) types of migrants. I wonder whether the authors asked some open-ended questions about what types of migrants (or general thoughts) the respondents had in mind after reading the frames. This may give us some hint about the type of migrant (and potential biases/associations) people had in mind when reading the respective treatment text. At the least, this shortcoming should be addressed in greater detail in the manuscript.

I have two concerns with the way that the outcome variable was formulated. First, the reference “a typical family spent about 1,500 takas on medicine last month” may have exacerbated social desirability bias, since it conveys the information that compared to what a typical family spends on healthcare, 100 takas are not much. Second, the way that this question is formulated, I wonder whether it may also have tapped into people’s support for NGOs more generally rather than just capturing people’s support for the specific migrant group described in the treatment text.

As an additional comment, I am curious why the authors did not use a continuous variable by asking respondents how much of the 100 takas they’d be willing to donate.

The authors discuss how much the average Bangladeshi household spends on healthcare. Is this the same amount that the average slum resident in Korail spends on healthcare? In general, I would like to hear more about in what way the survey site (the Korail slum) differs from other slums, or the country.

7. PLOS authors have the option to publish the peer review history of their article (what does this mean?). If published, this will include your full peer review and any attached files.

Reviewer #2: No

Reviewer #3: No

---

## [Author Response · Author response to Decision Letter 1]

7 Mar 2021

Rebuttal Memo

Willingness to help climate migrants:

A survey experiment in the Korail Slum of Dhaka, Bangladesh

PONE-D-20-35549R1

Reviewer #2

1. This is a strong revision and I would be delighted to see this research published, pending a few minor revisions listed below. I paid particular attention to the author(s)’ responses to the prior referee reports, and to their alterations of the manuscript to reflect their adjustments to critiques. In particular, the theoretical framework in the revised manuscript is considerably strengthened, the link between theory and empirics is more focused, and the empirical evidence remains robust to the different sensitivity checks that the author(s) have now introduced. Overall, I am impressed by the constructive revisions undertaken by the author(s) and see the author(s) as admirably responsive to the prior concerns raised in review.

Response:

Thank you.

Minor revisions:

1. In l.106 “especially if it involves citizens to incur private costs” would read better as “especially if it involves citizens incurring private costs.”

Response:

Done.

2. In response to R1’s excellent comments, the author(s) now address the issue of religious similarities and differences between the migrants and respondents quite extensively in the manuscript, noting that the Rohingya share the same Islamic faith as the majority of Bangladeshis (e.g., p.10), that respondents would likely not have associated the migrants with religious minorities like Hindus (p.12), and that respondents could have been expected to be “sympathetic to their co-religionists” (p.14, l.304). In neighboring India, which is frequently referenced in the manuscript, existing public opinion work shows that city residents belonging to the majority religious group (in the Indian case, Hindus) do not discriminate again rural-to-urban migrants based on religious profiles. Instead, residents from the dominant religious group appear to care mainly about the economic impact of migration. This is in fact consistent with what the author(s) find in the Bangladesh context: slum residents do not appear to heed religious concerns in responding to migration. The manuscript would be strengthened by citing and referencing these points as it would link the manuscript to broader debates in comparative politics.

Citation: Gaikwad, N. and Nellis, G. (2017), “The Majority‐Minority Divide in Attitudes toward Internal Migration: Evidence from Mumbai.” American Journal of Political Science, 61: 456-472.

Response:

Thanks for suggesting this article which we have referenced in the revised paper.

3. In l.343, the sentence is missing a closed parenthesis mark.

Response:

Corrected.

Overall, I continue to believe that this manuscript tackles a very important subject, and I expect it to foreshadow new work on climate change and migration.

Response:

Thank you.

Reviewer #3

1. This paper examines individuals’ willingness to provide assistance to climate migrants as compared to other types of migrants by means of a survey-embedded experiment conducted in a slum in Dhaka. The paper is well-written and addresses an important and very timely question. I believe that this study can make a contribution to the literature after addressing the issues below.

Response:

Thank you.

Theory

2. I think the authors need to provide a more specific discussion of the mechanism(s) underlying their main hypothesis, especially “H1: Survey respondents will be more willing to support climate migrants in relation to generic migrants.” The arguments that the authors provide (line 222: “extensive focus on climate change in media and the strong advocacy by the Bangladesh government in global forums”) are based on the assumption that citizens of Korail slum take their cues from the media and policymakers and will formulate their willingness to support (climate) migrants accordingly. However, this not only requires that slum residents have access to these specific information, but are also well-informed about what their national politicians advocate in global policy arenas. 

Response:

In this survey, we did not ask directly about how respondents learn about policy issues in general or about government’s positions on climate migrants. This is for two reasons. First, people typically get much of their policy information from different types of mass and social media. Of course, we do not know if this information is authoritative and if they comprehend this information. Indeed, we are not making any claim on how well informed or poorly informed individuals are about climate change, migration, or any national policies. We recognize that as boundedly rational actors, individuals develop opinions about issues based on incomplete information. Yet, no matter how incomplete or poorly informed individuals are, public opinion matters. And this is what we examine in the context for public support for a charity that provides health services to climate migrants and Rohingyas (in relation to generic migrants). 

Because mass and social media plays a crucial role in shaping public opinion, we have asked questions about respondent’s exposure to different types of media (Q15 of the survey):

a. Read a daily newspaper? 

b. Watch the news or a news program on television?

c. Listen to any news on radio or FM radio?

d. Read/watch news on your phone (e.g., Facebook, online portal etc.) or computer? 

We have analyzed if exposure to different media might influence respondent’s support for climate migrants or Rohingyas in relation to generic migrants. Anticipating some of concerns of the reviewer, we have not subsumed all types of media in category: we have examined them separately. The reason is that some respondents might rely on newspapers, and others on Facebook, to access policy information. Also, it is possible that some media might cover climate issues more extensively or effectively than others. For example, television might provide more content about challenges faced by climate migrants in relation to, say radio. Or, the television footage might create more empathy for climate migrants. If so, those with higher exposure to television might reveal higher support for the charity supporting climate migrants. 

Also, please note that our model controls for factors such as prior experience with floods that might make respondents more (or less) prone to access or pay attention to specific type of climate information in any media. Finally, because this is a survey experiment in which respondents are randomly assigned to different frames, unobserved heterogeneity in respondents’ characteristics should not influence support for any frame.

When we rerun the analysis for each media type separately, our results do not change (i.e. support for climate migrants or Rohingyas in relation to the generic frame). This gives us additional confidence that the medium through which respondents might receive information is not changing the support for climate migrants in relation to generic migrant. We have included this discussion in the revised paper.

 

3. I think the authors could expend more efforts to provide more micro-level arguments as to why they think climate migrants might be perceived in more favorable lights than other (generic) migrant types.

Response:

Fair point. The subject of public support for climate migration (in relation to other types of migration) is relatively new in the climate policy literature (although as we point out in the paper, there is extensive literature on levels of climate migration in parts of the world, and whether climate migrants should be recognized as refugees). We have identified a recent paper which investigates whether respondents might perceive climate migrants differently from other migrants. In the context of Germany, Hebling (2020) reports that German survey respondents are more supportive of climate change migrants, in relation to economic migrants. We have referenced this paper in the revised manuscript. 

Helbling, M. (2020). Attitudes towards climate change migrants. Climatic Change, 1-14.

3. My main issue with this manuscript is the design of the generic frame. The problem is that the lack of a specific driver of migration in this frame introduces the possibility for survey respondents to think about all sorts of migrant type, including climate migrants as well as both domestic and international migrants. I think this heterogeneity, but more importantly, the impossibility to identify what type of migrant the respondent is thinking about when answering whether they’d like to help that group of migrants, makes it very hard to interpret the outcome variable itself. What is the level of support we are benchmarking support for climate migrants against? We know from the literature that different types of migrants are seen in very different ways (as this manuscript also argues). Therefore, it is important to know how individuals perceive climate migrants compared to other (specific) types of migrants. I wonder whether the authors asked some open-ended questions about what types of migrants (or general thoughts) the respondents had in mind after reading the frames. This may give us some hint about the type of migrant (and potential biases/associations) people had in mind when reading the respective treatment text. At the least, this shortcoming should be addressed in greater detail in the manuscript.

Response:

Migrants have different characteristics. The vast literature on migration studies has examined public support when specific characteristics of the emigrant such as religion, gender, skill level, etc. are highlighted. We contribute to this literature by focusing attention to a specific characteristic that the literature has overlooked: climate change as a migration driver. Thus, the generic frame does not highlight any migration driver unlike the two treatment frames. Consequently, this research design allows us to assess the change in public support when one specific migrant characteristic (migration driver: climate change or religious persecution) is highlighted in the two treatment frames while all other information remains the same as the generic frame. 

This is also why we do not have any open-ended questions to investigate what types of migrants the respondents had in mind after reading the generic frame because we are not examining how respondents perceive generic migrants. Rather, we want to see how support for migrants might shift (in relation to the generic migrant) when one specific migration driver is highlighted. We have included a discussion on this subject in the manuscript.

4. I have two concerns with the way that the outcome variable was formulated. First, the reference “a typical family spent about 1,500 takas on medicine last month” may have exacerbated social desirability bias, since it conveys the information that compared to what a typical family spends on healthcare, 100 takas are not much. Second, the way that this question is formulated, I wonder whether it may also have tapped into people’s support for NGOs more generally rather than just capturing people’s support for the specific migrant group described in the treatment text.

Response:

We have acknowledged the issue of social desirability bias in the previous revision. If there is a bias, because respondents are randomly assigned to different groups, it cannot explain the difference in support between the generic and treatment frames. 

Similarly, we do not have data to speculate whether respondents were responding to support for NGOs. All respondents were given the same information cues with only difference between the generic and treatment frames: drivers of migration. Again, because respondents were randomly assigned to different frames, how respondents interpreted the information cue is not important for this study.

5. As an additional comment, I am curious why the authors did not use a continuous variable by asking respondents how much of the 100 takas they’d be willing to donate.

Response:

We discussed this possibility of using some sort of a slider scale to ask respondents for their support for 100-taka donation on say 1-5 scale. Because this was an in-person survey (as opposed to an online one), the survey firm thought that the logistical issues will be difficult if we were to ask respondents to use the slider on the smartphone and might even be distracting. Hence, we decided to work with a 5-point Likert scale – which is consistent with most survey experiments in the climate policy field. In the Appendix, we also present our results using the OLS estimator, where the dependent variable is treated as continuous. We have noted this in the revised paper.

6. The authors discuss how much the average Bangladeshi household spends on healthcare. Is this the same amount that the average slum resident in Korail spends on healthcare? In general, I would like to hear more about in what way the survey site (the Korail slum) differs from other slums, or the country.

Response:

We do not have data on healthcare spending in Korail slum in relation to other slums. Also, the issue of level of healthcare expenditure is not critical to our research design. Across all frames, generic frame and the treatment frame, we had the same ask for donation, 100 takas. In the revised manuscript, we have noted that this research design pertains to only one “price point,” namely 100 takas. To further validate our study, future work could look at different price points. Also, our paper already acknowledges that this study should be replicated in other locations/slums to validate its findings.

---

## [Decision Letter · Decision Letter 2]

16 Mar 2021

Willingness to help climate migrants:  A survey experiment in the Korail Slum of Dhaka, Bangladesh

PONE-D-20-35549R2

Dear Dr. prakash,

I am pleased to inform you that your manuscript has been judged scientifically suitable for publication. Both reviewers are satisfied with the changes you made to respond to their concerns. I recommend that you still incorporate the minor suggestions from Reviewer 3. Your manuscript will then be formally accepted for publication once it meets all outstanding technical requirements.

Kind regards,

Bernhard Reinsberg, Ph.D

Academic Editor

PLOS ONE

Additional Editor Comments (optional):

Reviewers' comments:

Reviewer's Responses to Questions

**Comments to the Author**

1. If the authors have adequately addressed your comments raised in a previous round of review and you feel that this manuscript is now acceptable for publication, you may indicate that here to bypass the “Comments to the Author” section, enter your conflict of interest statement in the “Confidential to Editor” section, and submit your "Accept" recommendation.

Reviewer #2: All comments have been addressed

Reviewer #3: All comments have been addressed

2. Is the manuscript technically sound, and do the data support the conclusions?

Reviewer #2: Yes

Reviewer #3: Yes

3. Has the statistical analysis been performed appropriately and rigorously? 

Reviewer #2: Yes

Reviewer #3: Yes

4. Have the authors made all data underlying the findings in their manuscript fully available?

Reviewer #2: Yes

Reviewer #3: Yes

5. Is the manuscript presented in an intelligible fashion and written in standard English?

Reviewer #2: Yes

Reviewer #3: Yes

6. Review Comments to the Author

Reviewer #2: The authors have successfully addressed my earlier concerns and suggestions in the revised version of the paper.

Reviewer #3: I'm pleased to see this revised version of the manuscript. I'm also glad to see that the authors had taken my suggestions and questions seriously and have sought to address most of the points that I have raised. With regards to point 3, however, I remain concerned about the formulation of the generic frame (and the non mutual exclusiveness of the treatment and generic frame), but also appreciate the fact that this the survey experiment has already been conducted and that there is no room for changes of the experimental design at this stage. I would have, however, welcomed a more critical self-reflection in the discussion.

Another minor point: In addition to the Helbling study the authors have identified, there is another study on the perception of environmental migrants in Vietnam and Kenya (Spilker et al;. 2020: Attitudes of urban residents towards environmental migration in Kenya and Vietnam), which might be even more relevant to this study given the geographical scope of the paper.

7. PLOS authors have the option to publish the peer review history of their article (what does this mean?). If published, this will include your full peer review and any attached files.

Reviewer #2: No

Reviewer #3: No

---

## [Editor Report · Acceptance letter]

6 Apr 2021

PONE-D-20-35549R2 

Willingness to help climate migrants: A survey experiment in the Korail Slum of Dhaka, Bangladesh 

Dear Dr. prakash:

I'm pleased to inform you that your manuscript has been deemed suitable for publication in PLOS ONE. Congratulations! Your manuscript is now with our production department. 

Kind regards, 

on behalf of

Dr. Bernhard Reinsberg 

Academic Editor

PLOS ONE